# `MatSwarm`: trusted swarm transfer learning driven materials computation for secure big data sharing

Ran Wang ®[1,2,3], Cheng Xu ®[1,2,4] ✉, Shuhao Zhang[3], Fangwen Ye ®[1], Yusen Tang[1], Sisui Tang ®[1], Hangning Zhang ®[1], Wendi Du[1] & Xiaotong Zhang ®[1,2,4] ✉

The rapid advancement of Industry 4.0 necessitates close collaboration among material research institutions to accelerate the development of novel materials. However, multi-institutional cooperation faces significant challenges in protecting sensitive data, leading to data silos. Additionally, the heterogeneous and non-independent and identically distributed (non-i.i.d.) nature of material data hinders model accuracy and generalization in collaborative computing. In this paper, we introduce the `MatSwarm` framework, built on swarm learning, which integrates federated learning with blockchain technology. `MatSwarm` features two key innovations: a swarm transfer learning method with a regularization term to enhance the alignment of local model parameters, and the use of Trusted Execution Environments (TEE) with Intel SGX for heightened security. These advancements significantly enhance accuracy, generalization, and ensure data confidentiality throughout the model training and aggregation processes. Implemented within the National Material Data Management and Services (NMDMS) platform, `MatSwarm` has successfully aggregated over 14 million material data entries from more than thirty research institutions across China. The framework has demonstrated superior accuracy and generalization compared to models trained independently by individual institutions.

The integration of Industrial Internet of Things (IIoT) and machine learning is revolutionizing research and development in materials science[1,2]. The advent of Industry 4.0 has revolutionized materials science through the integration of IIoT. Advanced sensors and data acquisition technologies enable real-time monitoring of material parameters such as temperature, hardness, melting point, and boiling point, providing unprecedented data support[3]. Concurrently, machine learning algorithms analyze this vast data, allowing researchers to predict material properties, optimize designs, and discover new materials based on performance, structural properties, and preparatory conditions[4,5].

However, creating accurate predictive models requires large, diverse training datasets. Today, various materials and big data platforms[6–8] have been developed, providing researchers with aggregated data. Nonetheless, for sensitive datasets that cannot be publicly shared, material data mining and analysis remain limited due to small sample sizes[9,10]. This poses a challenge for training effective models. While data augmentation[11,12] offers a potential solution, relying on simulated data can compromise model accuracy and generalization[13,14]. Additionally, even with sufficient samples, standardized testing environments and methodologies can limit data

[1]School of Computer and Communication Engineering, University of Science and Technology Beijing, 100083 Beijing, China. [2]Beijing Advanced Innovation Center for Materials Genome Engineering, University of Science and Technology Beijing, 100083 Beijing, China. [3]College of Computing and Data Science, Nanyang Technological University, 639798 Singapore, Singapore. [4]Shunde Innovation School, University of Science and Technology Beijing, 528399 Guangdong, China. ✉e-mail: xucheng@ustb.edu.cn; zxt@ies.ustb.edu.cn

diversity, further hindering model generalization for new materials. Transfer learning[15] is often used as a solution, but it involves sharing complete models with third parties, which raises concerns about data security and potential leakage.

To accelerate the development of new materials, a secure and collaborative computing methodology is essential. This approach must ensure data protection while allowing collaborative modeling across different organizations to improve model accuracy and generalization. federated learning (FL)[16] offers a viable solution by enabling organizations to collaborate without revealing their original data, sharing only insights from local models. This protects sensitive data while allowing effective aggregation[17]. However, the traditional FL framework, which relies on a central server to aggregate local model parameters[18], raises concerns about the integrity and authenticity of the global model[19]. This centralization also makes the server susceptible to internal and external attacks[20,21].

Moreover, most existing FL solutions have primarily been validated theoretically, using publicly available datasets and focusing on classification problems[22,23]. This theoretical focus fails to address the practical challenges faced by non-i.i.d. datasets owned by different organizations, where issues of model accuracy and generalization are more pronounced. The lack of empirical validation in real-world applications further questions the practicality and feasibility of these solutions. To truly harness the potential of FL in materials science, it is crucial to develop methodologies that not only perform well in controlled, theoretical settings but also demonstrate robustness and effectiveness in diverse, real-world environments. This will ensure the models are reliable, secure, and capable of advancing material discovery and development.

To accelerate materials science research and development, building on the Materials Genome Engineering (MGE) project[24], we developed the NMDMS platform[25,26] to facilitate the collection, storage, retrieval, and computation of material data. As the cornerstone of MGE's data applications, NMDMS platform provides data consumers with access to an extensive repository of material data contributed by over thirty research institutions across China. This platform also serves as a data exchange and sharing hub for materials researchers. Although the NMDMS platform provides basic collaborative computing services, it lacks solutions for handling the inherent limitations of FL in the context of material science. For example, while it achieves relatively high prediction accuracy for i.i.d. (independent and identically distributed) training sets, it falls short in generalization capability for non-i.i.d. training sets and cannot ensure the confidentiality and integrity of parameters during the training process.

Here, we introduce `MatSwarm` as part of the NMDMS platform to address the limitations in materials science collaboration, particularly in the context of Industry 4.0, where efficient cooperation among research institutions is crucial for accelerating novel material development. `MatSwarm` tackles the challenges posed by non-i.i.d. data and ensures the confidentiality and integrity of sensitive material information through a decentralized collaborative computing framework. To the best of our knowledge, this application of the `MatSwarm` framework is unprecedented in the materials field. Validated with real datasets from NMDMS, `MatSwarm` significantly enhances model training accuracy and generalization under heterogeneous data conditions. Additionally, by integrating trusted execution environments (TEE) based on Intel SGX, the framework ensures secure and accurate model aggregations. Ultimately, `MatSwarm` not only addresses the collaborative computing challenges but also unlocks the full potential of material data, driving innovation and meeting the demands of high-throughput computing and experimentation, thus accelerating material discovery. A general introduction on `MatSwarm` is available in Supplementary Movie 1.

## Results

To date, the `MatSwarm` platform for material genome engineering (MGE) has collected over 14 million pieces of valid material data[27]. The platform predominantly encompasses data on special alloys, material thermodynamics/kinetics, composite functional materials, catalytic materials, first-principles calculations, and biomedical materials. Data consumers from various fields can submit sharing tasks via the framework according to their specific needs, enabling collaborative prediction of material properties and the development of new materials with other stakeholders. In our experiments, we utilize the prediction of perovskite formation energies as an illustrative example to evaluate the performance of the `MatSwarm` framework. The following research questions (RQs) are addressed:

RQ 1: How does `MatSwarm` address security issues during the collaborative computing process in the material science domain?

RQ 2: What are the advantages of `MatSwarm` compared to other methodologies within the `MatSwarm`?

RQ 3: How do different factors affect the performance of `MatSwarm`, such as data distribution (non-i.i.d. vs i.i.d.), different local models and aggregation methods, and TEE?

RQ 4: How scalable is `MatSwarm` in terms of its performance, including the size of the dataset, the number of features, and the number of participants?

### Experimental setup

In this experiment, all services and participants' applications were deployed on cloud servers. The 16-core Intel Xeon (Ice Lake) Platinum 8369B processor with 32GB RAM (16GB as trusted RAM) was used to enable Intel ®Software Guard Extensions, allowing organizations to employ enclaves for protecting the confidentiality and integrity of their code and data. The `MatSwarm` framework was implemented on a consortium blockchain based on Hyperledger Fabric, with each node initiated as a Docker container and connected to the blockchain network using Docker Swarm[28]. Local models and aggregation methods are available for participants to choose from on the `MatSwarm` platform. The batch size was set to 128, the number of iterations was 200, and the learning rate was 0.002. In the training objective, $\gamma$ and $\lambda$ were set to 0.5 and 1, respectively.

Dataset and Model Selection. In our experiments, we illustrate our approach using the prediction of perovskite formation energies as a case study. We utilized perovskite data from our NMDMS platform to evaluate the performance of the `MatSwarm` framework, selecting 4016 perovskite samples. The training set consists of 3694 samples, evenly distributed among organizations. The test set comprises 322 samples. Detailed feature engineering on the dataset is described in Supplementary Note 4. Unless specified otherwise, the number of participants in the experiment is set to three. This experiment aimed to test the performance of `MatSwarm` for non-i.i.d. material data. To this end, we divided the training dataset into non-independent and identically distributed (non-i.i.d.) and independent and identically distributed (i.i.d.) datasets for comparative testing. For the non-i.i.d. dataset, since the label values are normally distributed, we divided the training set into three datasets with different means and variances. The distribution of label values in these datasets is illustrated in Supplementary Fig. 10.

Regarding model selection, unless otherwise specified, the local training models utilize a Multilayer Perceptron (MLP) neural network for training, with a hidden size of 32 and three network layers. On the `MatSwarm` framework, the task issuer can select different local training models and aggregation methods based on the sharing task. For joint training, all organizations' data was combined, and model training was also conducted using the MLP neural network.

Evaluated Attacks. In this scenario, four nodes participate in the collaborative task, with one acting as a Byzantine node launching the

attack. Since all attack methods target the gradients, we modify the model updates in this experiment to gradients instead of the model parameters. The aggregation methods include the five provided by the `MatSwarm` framework. We evaluate the impact of different attack methods on the accuracy of these aggregation methods both inside and outside the TEE. Given the susceptibility of existing swarm learning frameworks to data poisoning attacks[29], our experiment aims to demonstrate the robustness of `MatSwarm` against such attacks. We consider the following popular poisoning attacks:

- *Noise Attack.* The Byzantine nodes send noise-perturbed gradients generated by adding Gaussian noise to the honest gradients[30]. We set the Gaussian distribution parameters as $\mathbb{N}(0.1, 0.1)$.
- *Label-Flipping.* The Byzantine nodes flip the local sample labels during the training process to generate faulty gradients[31]. Specifically, a label $l$ is flipped as $-l$, where $l$ is the formation energy of perovskite in our experiment.
- *Sign-Flipping.* During each round of learning, participants calculate the gradients $\nabla f_{\theta}$ of the local model, which are then uploaded to a central server for aggregation[32]. After calculating the local gradients, the Byzantine nodes flip the signs of these gradients and send $-\nabla f_{\theta}$.
- *A Little is Enough.* The Byzantine nodes send malicious gradient vector with elements crafted[33]. For each node $i \in [d]$, where $d$ is the number of Byzantine nodes, the Byzantine nodes calculate mean ($\mu_i$) and standard deviation ($\sigma_i$) over benign updates, and set corrupted updates $\Delta_i$ to values in the range $(\mu_i - z_{max}\sigma_i, \mu_i + z_{max}\sigma_i)$, where $z_{max}$ ranges from 0 to 1. We set $z_{max} = 0.3$ in our experiment.
- *Inner Product Manipulation.* The primary goal of IPM is to disrupt model performance by manipulating the inner product of gradients to affect the direction and speed of model training[34]. For example, an attacker could enhance or diminish the effects of gradients in a particular dimension. We set the scaling factor $\alpha = 2$, the gradient mean to be $\overline{\nabla f_{\theta}}$, and the gradient of the attack sent to be $-\overline{\nabla f_{\theta}} * \alpha$.

### Security analysis (RQ1)

Confidentiality protection for local datasets: this framework enables collaborative computing among multiple organizations while maintaining the confidentiality of local datasets. Traditional centralized machine learning methods require storing all datasets on a central server, posing risks of sensitive data leakage. Through `MatSwarm`, each organization trains models on its local dataset without sharing the original data. Instead, organizations only share encrypted model parameters, not raw data. This approach prevents the disclosure of sensitive information without disrupting the task processes of the participating organizations.

Secure model training and aggregation based on TEE: ensuring the security of model training processes during swarm learning is a significant challenge. To address this issue, `MatSwarm` employs a TEE established by Intel SGX. In this framework, the original dataset is encrypted before being sent to the blockchain, using a shared key established through the Diffie-Hellman key exchange protocol. This ensures that data cannot be stolen or tampered with during transmission. During model training and aggregation, the SGX Enclave performs these operations in a trusted execution environment, preventing attackers from accessing or altering model parameters.

Blockchain-based secure storage: this framework uses blockchain technology to replace untrusted third parties, significantly reducing the risk of data leakage. Smart contracts are employed to standardize and automate model training and aggregation processes. Transactions are stored on the blockchain in hash form, and due to the uniqueness of hash values, any tampering with transaction data will result in a change in the hash value. During the consensus process, nodes reject transactions with inconsistent hash values, ensuring the integrity of

global model storage and preventing network attacks. Additionally, digital signatures and hashes protect model updates, further enhancing the security of model training and preventing tampering or contamination.

Impact of attacks on different aggregation methods inside/outside TEE: as shown in Fig. 1a-e present test results in a non-TEE environment. The results indicate that different aggregation methods, by design, can resist various data poisoning attacks. However, no single aggregation method can resist all types of data poisoning attacks. The convergence speed and final model accuracy are affected to varying degrees depending on the specific attack and aggregation method used. To verify the TEE's resistance to data poisoning attacks, we tested the aggregation methods that were ineffective in the non-TEE within the TEE. Figure 1f-j show that the TEE effectively resists all types of data poisoning attacks. The convergence speed and model accuracy remain virtually unaffected, closely matching the performance observed in the absence of attacks. This demonstrates that `MatSwarm` can effectively mitigate the risk of data poisoning attacks.

### Methodologies comparison (RQ2)

Within the `MatSwarm` framework, we conducted comparative experiments on prediction accuracy and response time between `MatSwarm`, local independent training (referred to as "Solo"), joint data training (referred to as "Joint"), and other existing solutions, including FedAvg[35], FedProx[36], Homomorphic Encryption Federated Transfer Learning (referred to as "HE-FTM")[37], and a similar framework proposed by Kalapaaking et al. (referred to as "Trust-FL")[38], to illustrate the performance advantages of this framework. The performance comparison between `MatSwarm` and other methodologies is presented in Fig. 2.

The results of model accuracy, evaluated using mean squared error (MSE), are shown in Fig. 2. `MatSwarm` significantly improves prediction accuracy compared to Solo while maintaining the privacy of local datasets across various organizations. Among the methodologies compared, `MatSwarm` achieves prediction accuracy closest to that of Joint, which can be considered the upper bound of accuracy for collaborative computation. In contrast, HE-FTM involves polynomial approximation for evaluating nonlinear functions, resulting in some accuracy loss during training. Trust-FL, employing a horizontal FL model, is more suited for i.i.d. training data and is less effective at predicting non-i.i.d. material data models. In terms of prediction accuracy, our model is more suitable for the material science domain, demonstrating better prediction accuracy.

Regarding response time, as shown in Fig. 2, Solo exhibits the shortest response time for each organization. Due to the communication required between organizations, `MatSwarm` takes longer to execute compared to Solo and Joint. The average response time of `MatSwarm` increases by approximately 4 seconds compared to Joint. Despite this increase, the security and privacy protection offered by `MatSwarm` are highly valuable. Moreover, in practical applications, organizations typically do not require real-time model training, and the response time difference remains within an acceptable range. Notably, compared to HE-FTM, our model demonstrates lower computational complexity and significantly improved response time. Compared to Trust-FL, our framework shows a slight increase in response time, primarily due to the enhanced security measures. Model training in our framework occurs in a trusted execution environment, adding some communication overhead. Additionally, the blockchain consensus algorithm, inspired by PBFT, effectively addresses security concerns arising from Byzantine nodes. Although the consensus algorithm slightly impacts response time by increasing communication frequency, the trade-off is justified by the improved security performance.

### Ablation experiment (RQ3)

To understand how different factors affect the performance of `MatSwarm`, we conducted ablation experiments varying data label

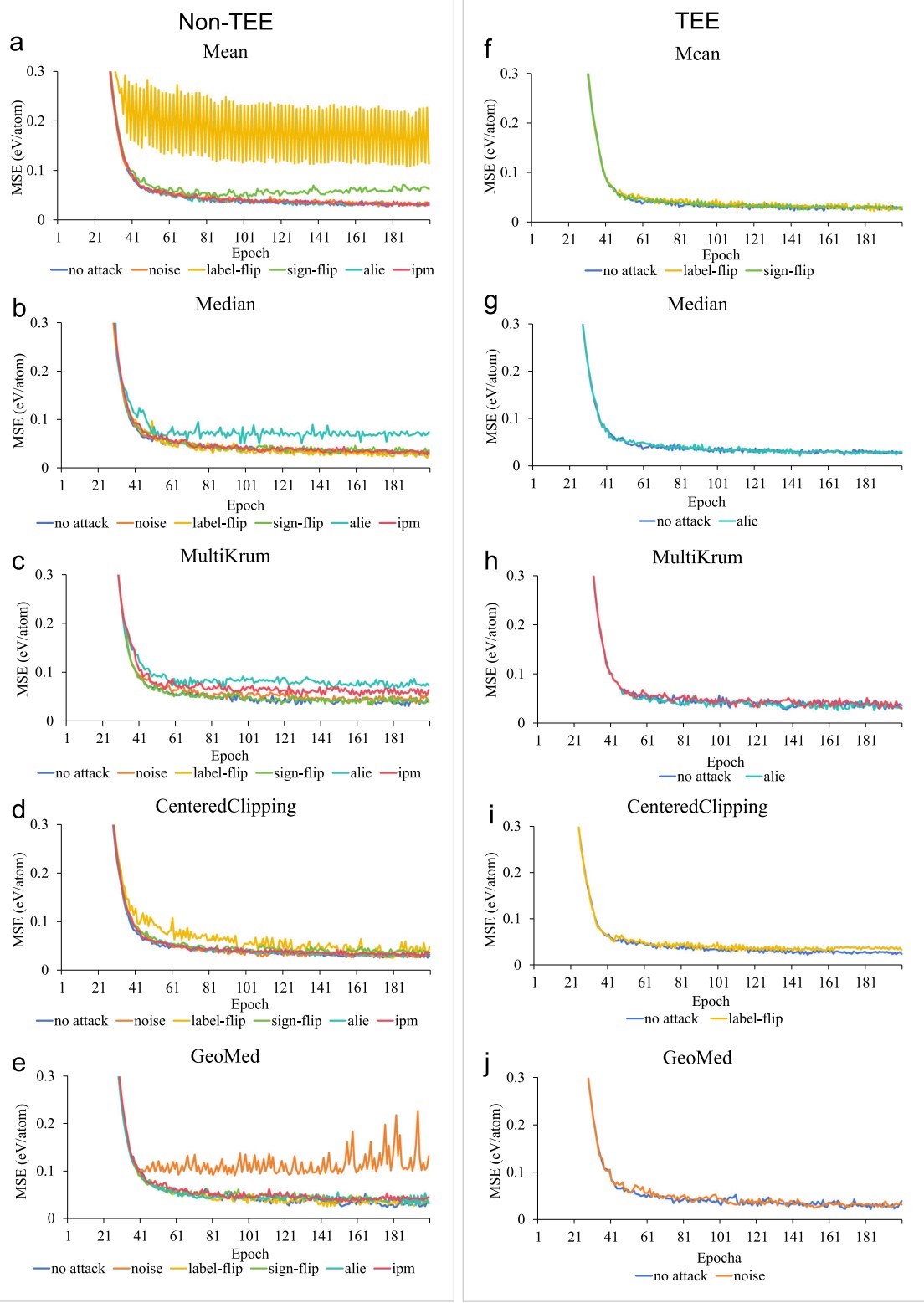

**Fig. 1 | Comparison of the mean squared error (MSE) across different aggregation methods under various attack scenarios.** Here, MSE indicates the mean squared error between the predicted and label values of perovskite formation energy. The total number of samples in the training dataset ($n = 3694$) is evenly distributed among the organizations, and the test set consists of 322 samples. The X axis represents the number of training rounds, while the Y axis denotes prediction accuracy measured by the MSE value. **a**–**e** display results tested in a non-TEE (Non-Trusted Execution Environment), whereas **f**–**j** present results tested in a TEE (Trusted Execution Environment), specifically within the MatSwarm framework, utilizing various global aggregation methods. Source data are provided as a Source Data file.

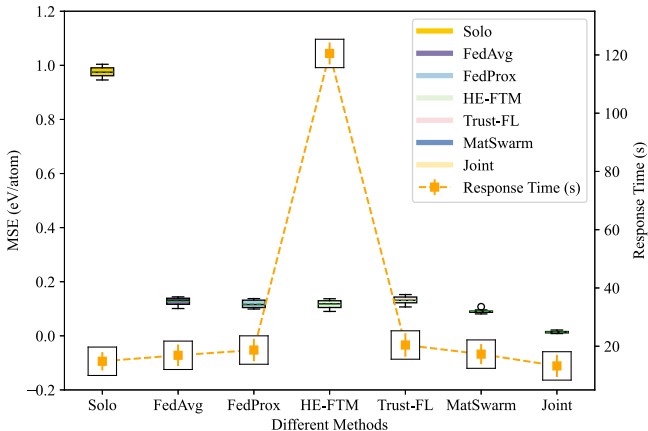

**Fig. 2 | Performance comparison between** MatSwarm **and other methodologies across three organizations.** The total number of samples in the training dataset ($n = 3694$) is evenly distributed among the organizations, and the test set consists of 322 samples. The methods evaluated include Solo (individual training), FedAvg, FedProx, HE-FTM (Homomorphic Encryption-based Federated Transfer Learning), Trust-FL (Trusted Execution Environment-based Federated Learning), MatSwarm, and Joint (centralized training). The left *y* axis represents the mean squared error (MSE, eV/atom), indicating prediction accuracy, and the right y-axis shows the Response Time (seconds) for each methodology. The boxplots depict the MSE distribution over 10 experimental runs, showing the mean, quartiles, and outliers, while the line plot with error bars indicates the average response time. To enhance the visibility of the standard deviation, a zoom effect is applied in the inset black boxes, emphasizing the variability. Source data are provided as a Source Data file.

distribution, local model architectures, and w/wo a TEE. Unless specified otherwise, all local models use the same MLP architecture, and the aggregation algorithm is Mean.

1) non-i.i.d. VS i.i.d.: To demonstrate the performance of the MatSwarm on non-i.i.d. material data, we tested both non-i.i.d. and i.i.d. datasets.

- *i.i.d. Training Sets*: Fig. 3a depicts the prediction results for perovskite formation energy using i.i.d. training sets selected independently for each organization. Our algorithm exhibits extremely high prediction accuracy for the i.i.d. dataset, nearing the accuracy of Joint.
- *Non-i.i.d. Training Sets*: As shown in Fig. 3b, the prediction accuracy for the non-i.i.d. dataset is slightly lower compared to the i.i.d. dataset but still close to the accuracy of Joint. Compared to Solo, the accuracy for Org1 decreases due to the different label distributions between its data and the test set. A similar trend is observed for Org3.

However, as displayed in Table 1, using MatSwarm for predictions, the prediction MSE for Organization 1 decreased from 1.0291 to 0.2096, and for Organization 3, it decreased from 1.6159 to 0.5849, with the global model achieving an accuracy as low as 0.0903. Despite Organization 2 having a similar label distribution to the test set and thus showing good prediction accuracy, its local model prediction accuracy also improved slightly after training with MatSwarm. This demonstrates that MatSwarm has strong generalization capabilities for non-i.i.d. material data.

2) Different local models and aggregation methods: Since MatSwarm will perform various training tasks beyond predicting perovskite formation energy, the choice of local models and aggregation methods significantly impacts the accuracy of model training for different tasks. In this experiment, we compared the performance of MatSwarm using different local models and aggregation methods to identify the most suitable collaborative computing scheme for predicting perovskite formation energy. The local models capable of solving regression

problems include MLP, recurrent neural network (RNN), Lasso, and long short-term memory (LSTM). The aggregation methods considered are Mean, Median, MultiKrum, CenteredClipping, and GeoMed.

Ultimately, we obtained the prediction results shown in Fig. 3c, d, and Table 1. The results indicate that using MLP within MatSwarm is the most suitable for predicting perovskite formation energy. Building on the MLP local model architecture, we tested the impact of different aggregation methods on model accuracy and response time. In terms of accuracy, Mean and CenteredClipping achieved the higher precision, while Mean was the most efficient in terms of response time. Therefore, to choose a suitable aggregation method, one should balance the trade-offs among the needs of efficiency, accuracy, and security to achieve an optimal solution. This modular development approach facilitates participants in selecting the most suitable solutions for training tasks and simplifies platform iterations and updates to meet diverse training demands in the material science domain.

3) non-TEE vs TEE: To evaluate the impact of TEE on the accuracy and efficiency of the MatSwarm framework, we compared the MSE and response time of MatSwarm before and after using TEE. The comparison, shown in Fig. 4, indicates that using TEE does not significantly affect the prediction accuracy of the model, whether training is conducted individually, with MatSwarm, or on joint data. However, the use of TEE introduces some communication overhead, leading to an increase in response time. In the materials science domain, unlike in transaction systems, there is typically no strong demand for real-time response, and large model training often takes hours. Therefore, the increase in response time due to TEE is negligible compared to the enhancement in security it provides. The TEE-based MatSwarm fully meets the performance requirements for model prediction in the materials science field.

## Scalability testing (RQ4)

In this experiment, we evaluated the scalability of MatSwarm by examining the impact of different dataset sizes, the number of features, and the number of participants. It is important to note that our NMDMS platform operates within a limited number of material organizations. Currently, the platform accommodates up to 30 registered material organizations, with typically no >10 participants in a sharing task. Therefore, in our experiments, we tested the framework with a maximum of 15 participants (material organizations).

1) Dataset size: Fig. 5a illustrates the MSE and response time of MatSwarm across varying dataset sizes. The results indicate that dataset size has a negligible impact on the response time of MatSwarm, while the model accuracy continues to improve with increasing amounts of data. Notably, even when each organization's dataset comprises only 30% of the original dataset, our method demonstrates high accuracy. This indicates that our approach can achieve highly accurate training models even with small sample sizes within each organization, effectively addressing the small sample problem in the materials science domain.

2) Number of features: as shown in Fig. 5b, increasing the number of features does not significantly affect the response time of MatSwarm, demonstrating good scalability in terms of computational efficiency. In terms of prediction accuracy, even with sample features constituting only 30% of the total features, our method achieves an MSE value as low as 0.155, indicating high accuracy. Therefore, using MatSwarm, even if each organization can only obtain a limited number of feature values, it is still possible to achieve highly accurate training models. This makes our approach particularly effective for scenarios where organizations have limited data or feature availability, ensuring robust and reliable model performance. Furthermore, after reaching ~90% of all features, the addition of less important features does not substantially impact accuracy. In practical applications, selecting an appropriate set of features is crucial for balancing accuracy and efficiency, often involving feature extraction optimization methods[39,40].

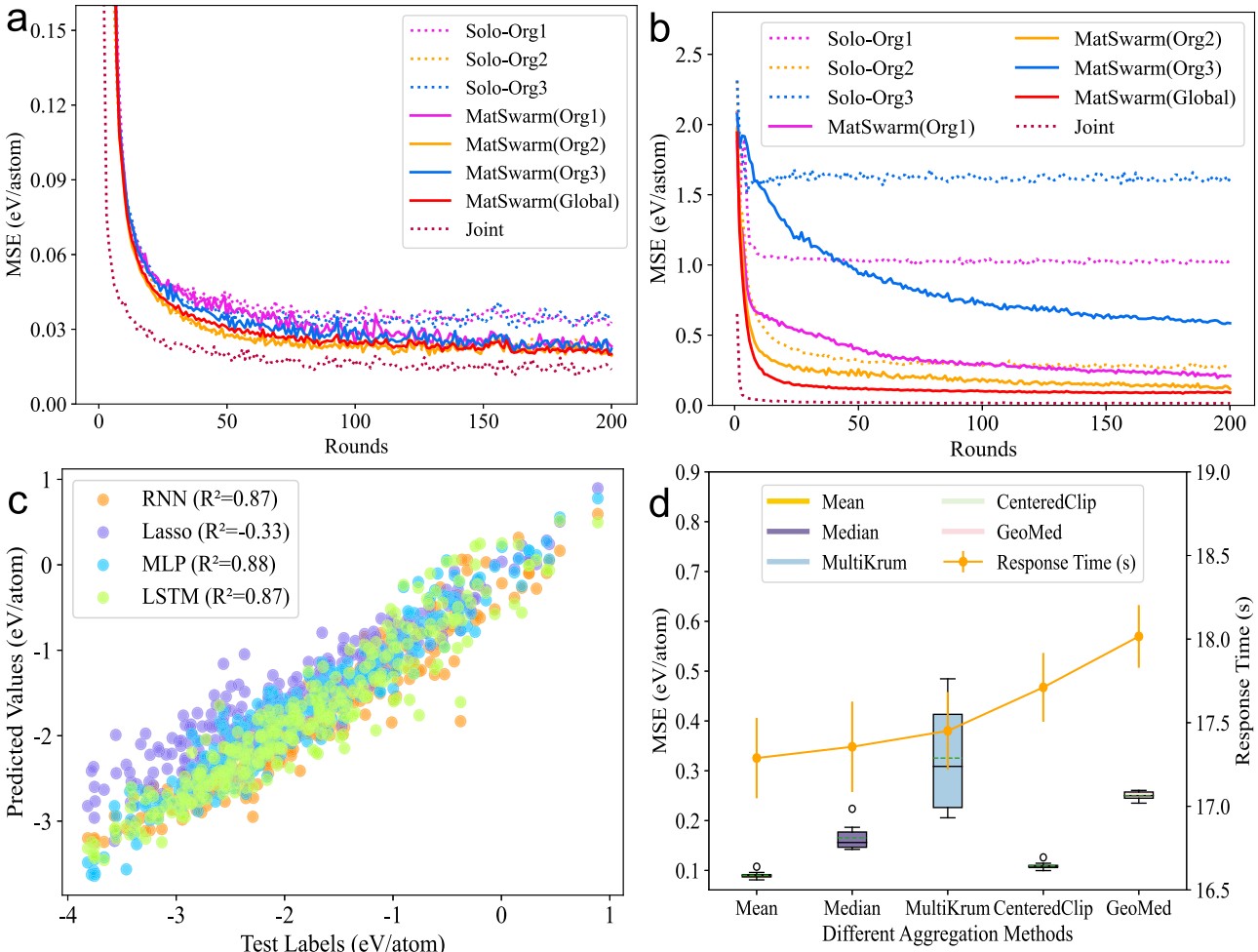

**Fig. 3 | Performance comparison on the influence of different data label distributions. a**, **b** respectively represent the mean squared error (MSE) and response time when predicting perovskite formation energy under iid (independent and identically distributed) and non-iid (non-independent and identically distributed) conditions. The models tested include the independently trained models by each organization (Solo-Org1, 2, 3), the local models under the `MatSwarm` framework (`MatSwarm` (Org1, 2, 3)), the global model under the MatSwarm framework (`MatSwarm` (Global)), and the model trained after aggregating all training datasets (Joint). **c** Scatter plot of prediction results under various local models. **d** Comparison of MSE and response time across different aggregation methods. The total number of samples in the training dataset ($n = 3694$) is evenly distributed among the organizations, and the test set consists of 322 samples. Source data are provided as a Source Data file.

3) Number of participants: as shown in Fig. 5c, the response time of `MatSwarm` increases linearly with the number of participants. This increase is primarily due to the additional time required for communication and data coordination. The observed increase in response time aligns with theoretical expectations. In terms of accuracy, the prediction accuracy of `MatSwarm` shows a notable upward trend as the number of participants increases. However, after a certain threshold, the accuracy may slightly decline due to issues such as communication delays, data inconsistencies, and model overfitting introduced by a higher number of participants. Therefore, in the participant selection process, more is not necessarily better. This demonstrates that `MatSwarm` can effectively learn the data characteristics of each organization, achieving highly accurate training models without the need for a large number of participants for collaborative training. Consequently, this approach can also enhance the efficiency of model training.

## Discussion
### Advantages of the `MatSwarm` framework
Security: `MatSwarm` incorporates advanced security measures to ensure data confidentiality and integrity. A key component of our security strategy is the use of TEEs, specifically Intel SGX, which protect code and data from external attacks during computation. This approach effectively mitigates poisoning attacks associated with traditional FL setups. Furthermore, our experimental setup included various attack scenarios to test the resilience of the `MatSwarm` framework. These tests demonstrated that `MatSwarm` effectively maintains data integrity and model accuracy, even in the face of malicious attempts to corrupt the training process.

Feasibility: `MatSwarm` is crucial for enabling collaborative computation over non-i.i.d. material data, a common challenge due to the diverse nature of data sources and formats in this field. Compared to independent training by organizations and other FL methodologies, our method significantly improves prediction accuracy and generalization ability. This highlights `MatSwarm`'s potential to unlock the full value of material data, facilitating more informed and accurate materials discovery and development processes. Extensive testing with real-world data from the material science domain validated the usability of the `MatSwarm` framework. By engaging with actual datasets from participating institutions, we demonstrated the feasibility and accuracy of the models generated through our platform. This use of real data underscores the framework's ability to address the 'data silos' problem prevalent in materials science.

**Table 1 | Mean squared errors (MSEs, eV/atom) for the prediction of perovskite formation energies using various local models**

| Local Model | RNN | Lasso | MLP | LSTM |
|---|---|---|---|---|
| Solo-Org1 | 0.9146 ± 0.1382 | 1.3946 ± 0.0298 | 1.0291 ± 0.0551 | 1.1483 ± 0.0845 |
| Solo-Org2 | 0.2681 ± 0.0169 | 0.4313 ± 0.0026 | 0.2746 ± 0.0233 | 0.3481 ± 0.0292 |
| Solo-Org3 | 1.4023 ± 0.0692 | 2.1218 ± 0.0029 | 1.6159 ± 0.0405 | 1.5687 ± 0.0586 |
| MatSwarm(Org1) | 0.2341 ± 0.0232 | 1.1829 ± 0.0045 | 0.2096 ± 0.0474 | 0.1388 ± 0.0302 |
| MatSwarm(Org2) | 0.2156 ± 0.0199 | 0.2862 ± 0.0032 | 0.1176 ± 0.0143 | 0.1291 ± 0.0273 |
| MatSwarm(Org3) | 0.3769 ± 0.0866 | 1.2222 ± 0.0044 | 0.5849 ± 0.0386 | 0.1987 ± 0.0759 |
| MatSwarm(Global) | 0.0869 ± 0.0148 | 0.2595 ± 0.0035 | 0.0903 ± 0.0073 | 0.0941 ± 0.0194 |
| Joint | 0.0147 ± 0.0021 | 0.0191 ± 0.0039 | 0.0138 ± 0.004 | 0.0125 ± 0.0024 |

The total number of samples in the training dataset (n = 3694) is evenly distributed among the organizations, and the test set consists of 322 samples. "Solo-Org1," "Solo-Org2," and "Solo-Org3" represent predictions made using only the local datasets from each organization, while "MatSwarm(Org1)," "MatSwarm(Org2)," "MatSwarm(Org3)," and "MatSwarm(Global)" correspond to the local and collaborative model predictions facilitated by MatSwarm. "Joint" denotes the results of joint training where all raw data are shared across organizations. Data are presented as mean values with the standard deviation (mean ± standard deviation). Source data are provided as a Source Data file.

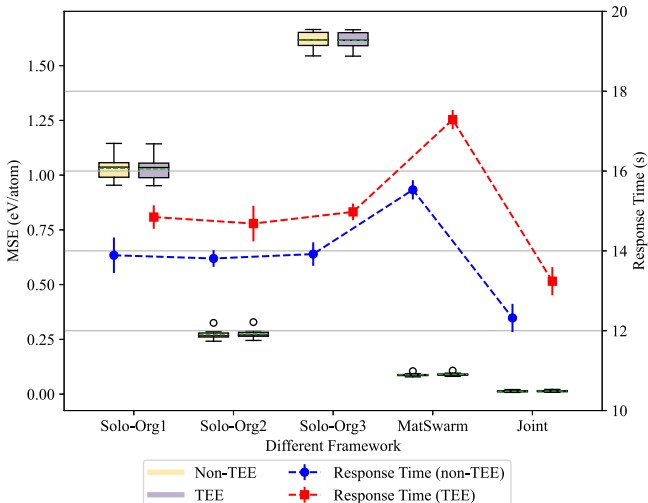

**Fig. 4 | Performance comparison of MatSwarm between trusted execution environment (TEE) and non-TEE conditions.** The figure presents a comparison of the Mean Squared Error (MSE) and response times for various frameworks under TEE and non-TEE settings across 10 experimental runs. The training dataset consists of 3694 samples, evenly distributed among the organizations, with a test set of 322 samples. The evaluated frameworks include Solo-Org1, Solo-Org2, Solo-Org3, MatSwarm, and Joint. Solo-Org1, Solo-Org2, and Solo-Org3 represent independent training approaches where each organization trains its model using only local data. The Joint framework involves centralized training where all datasets are combined into a single model. Source data are provided as a Source Data file.

**Scalability:** MatSwarm has been rigorously tested across multiple dimensions, including varying dataset sizes, feature quantities, and participant counts. The results show that the model maintains high and stable predictive accuracy, demonstrating excellent scalability and practical applicability. This consistent performance, even with smaller sample sizes and fewer features, underscores MatSwarm's capability to adapt to a broad range of scenarios. Such robustness enhances its potential for widespread adoption in collaborative settings that require handling complex, heterogeneous data landscapes. Additionally, the MatSwarm platform utilizes a modular architecture, allowing participants to select appropriate local models and aggregation methods based on their training tasks. As task demands increase, we will continuously iterate and update the platform's local models and aggregation methods. This approach aims to address various challenges in the material science domain, including performance prediction, material classification, and structural optimization, ultimately creating a versatile collaborative computing platform.

**Adaptability:** MatSwarm is a secure collaborative computing framework designed for non-public data across organizations on the NMDMS, specifically addressing key regression challenges in the materials science domain. In this paper, we demonstrate the capabilities of the MatSwarm framework by using it to predict perovskite formation energies, selecting a perovskite dataset as our example case. To be noted, our framework is suitable for general regression tasks within the materials science domain, such as predicting the elastic properties of silicon materials and optimizing the microstructure of high-performance alloys. For each shared task, participants can choose relevant datasets from their organization based on the task's requirements. This ensures that the framework is not restricted to specific datasets during implementation; instead, it dynamically adapts to select appropriate local datasets according to the specific needs of each task.

Moreover, although MatSwarm is specifically designed for collaborative computing in the materials science domain, its design principles can be leveraged by other domains with similar needs to construct their own swarm-based collaborative computing frameworks. For other domains with similar application requirements, the framework can be adapted by modifying the objective function and selecting suitable local models and aggregation methods to fit specific needs. Additionally, in Section 6 of the Supplementary Materials, we provide a detailed guide on how to extend and apply the MatSwarm framework to other domains.

### Limitations of the MatSwarm framework

Implementation complexity: while incorporating TEEs enhances security and privacy, it also increases the complexity of system setup and operations, necessitating robust infrastructure and specialized expertise. To mitigate this, we provide detailed platform deployment and configuration documentation in the supplementary materials, which stakeholders can use to deploy new training tasks on this platform.

Potential latency issues: the decentralized nature of blockchain and remote attestation based on TEEs can introduce delays in model training and aggregation. However, in the field of materials science, real-time requirements for training are not stringent. The minor increase in latency is negligible compared to the benefits of resolving the issue of data silos in material data.

Hardware dependency: dependence on TEEs, such as Intel SGX, to protect data during computation may limit the applicability of our framework in environments without such hardware support. Nevertheless, our demonstration system offers the option to choose whether to use TEEs to secure the confidentiality of the model aggregation process. Even without TEE protection, data security during transmission is ensured through data encryption and secure communication channels. In the future, we plan to offer additional privacy protection technologies, such as homomorphic encryption and differential privacy, to support a broader range of application scenarios.

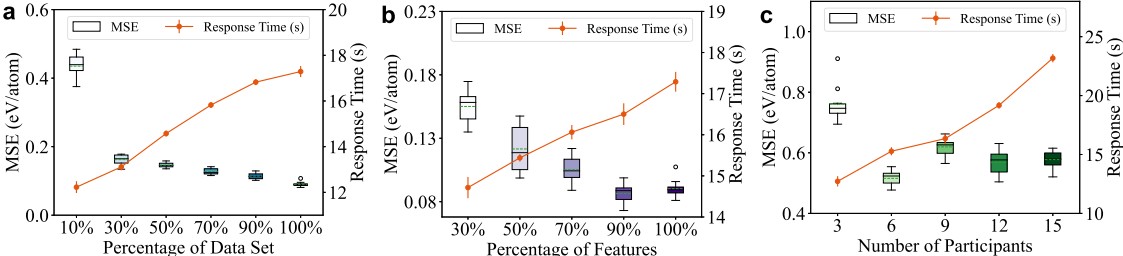

**Fig. 5 | Scalability testing of** `MatSwarm`. The boxplots depict the variability in Mean Squared Error (MSE) across 10 experimental runs, while the response time (in seconds) is represented by the orange line with error bars. **a** It illustrates the impact of dataset size on MSE and response time. The total training dataset consists of 3694 samples, evenly distributed among the organizations, with a test set of 322 samples. The $x$ axis represents different percentages of the dataset, ranging from 10% to 100%. As the dataset size increases, MSE (eV/atom) decreases, indicating improved model performance. **b** It examines the effect of feature selection on MSE and response time, with a total of 147 features; the $x$ axis shows the percentage of features used, from 30% to 100%. **c** It explores how the number of participants influences MSE and response time, with 3694 training samples evenly distributed among participants, ranging from 3 to 15, and a test set of 322 samples. Source data are provided in the Source Data file.

## Methods

### Overall architecture of `MatSwarm`

In this section, we present our proposed framework, `MatSwarm`, designed for the secure sharing of material big data using swarm transfer learning combined with TEEs. Table 2 summarizes the critical symbols used in our framework. The organizations illustrated in Fig. 6 represent examples of entities involved in materials science. It is noteworthy that our `MatSwarm` framework is primarily used to address challenges in collaborative computing within the domain of materials science, as evidenced by its application to a regression problem, such as predicting material properties like perovskite formation energies, as discussed in this paper. Nevertheless, the framework possesses the potential for extension and application in other fields facing analogous collaborative computing challenges. Further elaboration on this aspect can be found in Supplementary Note 5.

The `MatSwarm` framework enables collaborative computing tasks between material organizations. As depicted in Fig. 6, `MatSwarm` involves multiple organizations (denoted as $N$) collaborating to execute shared tasks. Each organization is responsible for training its own local models. The blockchain nodes provide a distributed computing environment for the participating organizations and store aggregated models. Additionally, the trusted execution environment ensures the secure aggregation of local model parameters and collaborates with the blockchain to generate the swarm global model.

1. Organizations: within the `MatSwarm` framework, organization $O_i$ $(1 \le i \le N)$ collaboratively trains models to meet shared material performance prediction requirements. Initially, each organization conducts material features sampling locally, and the collected samples are stored as local datasets on their respective cloud servers. Subsequently, organizations choose an appropriate machine learning method to train a local model. To ensure security during model training, each organization deploys at least one blockchain node on an Intel SGX-enabled cloud server, and the local model training is performed in SGX's application enclave. This setup establishes encrypted and authenticated channels, allowing sensitive data to be securely transferred between the cloud server and the Intel SGX Enclave.

2. Blockchain Network: `MatSwarm` leverages the decentralized nature of blockchain to create a collaborative computing environment. Each organization joins the blockchain network at local blockchain nodes. Within the `MatSwarm` framework, three transaction types are defined: retrieval, sharing, and uploading. The retrieval transaction verifies the existence of relevant sharing global models on the blockchain before initiating a new sharing task. The sharing transactions involve organizations initiating new tasks, such as material performance prediction, with the option for other organizations to participate. The uploading transactions store the final global model on the blockchain, ensuring its integrity and preventing tampering, thus facilitating model retrieval and usage by other organizations.

3. Trusted Execution Environment: the TEE, implemented via Intel SGX, ensures the confidentiality and integrity of local and global models. Each organization applies for two Application Enclaves (denoted as AE) in SGX. $AE_1$ is used to load encrypted local datasets and execute local models, ensuring confidentiality and integrity during execution. $AE_2$ is used to aggregate global models. This approach ensures the integrity of model aggregation, with all organizations automatically executing the same model aggregation code through smart contracts[41] in $AE_2$. Smart contracts automate the enforcement and management of agreed-upon processes and conditions, ensuring consistent execution, eliminating discrepancies, enhancing security, and reducing reliance on third-party intermediaries. Additionally, the Quoting Enclave (denoted as QE) generate attestation REPORT **R** to assist in remote authentication between AEs in various organizations.

## Table 2 | Notations

| Symbol | Definition |
|---|---|
| O | Organization |
| BN | Blockchain node |
| TEE | Trusted execution environment |
| AE | Application enclave |
| QE | Quoting enclave |
| **DL** | Local dataset |
| Kr | Symmetric encryption key |
| Er(. , Kr) | Symmetric encryption algorithm |
| $M_L$ | The parameter set of local model |
| $M_G$ | The parameter of global model |
| $(PK_{IAS}, PR_{IAS})$ | Public-private key pairs generated by Intel authentication services |
| **QG** | Quotation for global model |
| **R** | REPORT structure information |
| **M** | Enclave identity information |
| MAC | Message authentication code |
| **(AKp, AKr)** | (public, private) key pairs for authentication keys |
| Sign(. , AKr) | signature algorithm |
| verify(Sign(. , AKr), AKp) | Validation function |
| $[\cdot]_{PK_{IAS}}$ | Authenticated public key encryption |

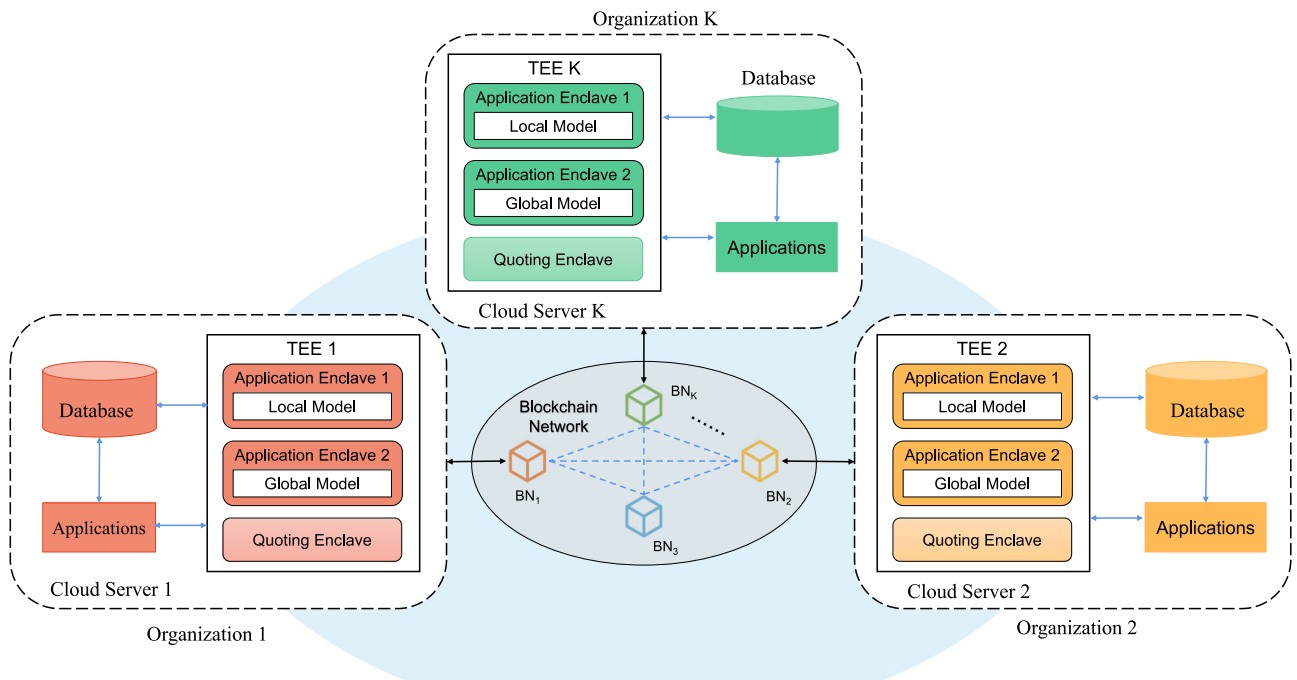

**Fig. 6 | The overall architecture of `MatSwarm` framework.** The `MatSwarm` framework facilitates collaborative material performance prediction across multiple organizations by ensuring secure local model training and global model aggregation. The blue dashed lines in the figure represent messages passing between blockchain nodes (BN). The blue solid lines represent messages passing between components within the organization's cloud server. The black solid lines represent messages passing between the local cloud server of each organization and its local blockchain node. Each organization utilizes its cloud server to store local datasets in a database, with applications managing data processing and model execution. Local models are trained within the trusted execution environment (TEE) in Application Enclave 1 through smart contracts, which automate and secure the execution process. The blockchain network underpins decentralized computing, securely storing and sharing the aggregated global models. Global model aggregation is performed in Application Enclave 2, ensuring data integrity and confidentiality. The quoting enclave within the TEE generates attestation reports to support remote authentication between enclaves across organizations.

## Problem formulation

We consider a `MatSwarm` framework constructed by $N(N > 2)$ organizations, where $K(K \leq N)$ organizations are in a sharing task, each possessing a local dataset $DL_i$, $\forall i \in K$. Each organization maintains a local model $f_{\theta_i} : \mathbf{X}_i \rightarrow y_{pre_i}$ with parameters $\theta_i$, where $\mathbf{X}_i$ and $y_{pre_i}$ denote the input and output spaces, respectively. In our study, we assume that all organizations have the same input/output specifications and homogeneous local model architectures. However, they may choose different local models and aggregation methods based on the sharing task. The objective is to collaboratively train the local models to ensure that each generalizes well on the joint data distribution, thereby improving prediction accuracy for non-i.d.d. material data.

To achieve this objective, we propose a swarm transfer learning method within the `MatSwarm` framework. The core of our method is to identify invariances between resource-rich source domains and resource-scarce target domains, facilitating the learning of common representation spaces and enabling knowledge transfer across domains. The objective function reveals that during the swarm transfer learning process between organization $O_i$ and organization $O_{i+1}$, local model training is interdependent, necessitating the exchange of intermediate training results. The training process adheres to a linear cycle method, with sequential training conducted between organizations in the order $[O_1 \rightarrow O_2 \rightarrow . . . \rightarrow O_K \rightarrow O_1]$. The completion of training between organizations $O_K$ and $O_1$ signifies the end of a local training round. After each round of local model parameter updates, the parameters are aggregated, and the updated global model parameters are sent back to each organization for the next round of local model updates. This iterative process continues until the model converges to a specified threshold.

The training objective is typically formulated as the following algorithm:

$$\min_{\theta_{i,i+1 \bmod K}} f(\mathbf{X}_{i,i+1 \bmod K}, y_{i \bmod K}) = \sum_{i=1}^{K} \mathscr{L}_1(\mathbf{X}_{i \bmod K}, y_{i \bmod K}) + \gamma \mathscr{L}_2(\mathbf{X}_{i,i+1 \bmod K})$$
$$+ \frac{\lambda}{2}(\| \theta_{i \bmod K}\|^2 + \| \theta_{i+1 \bmod K}\|^2) \quad (1)$$

$$\mathscr{L}_1(\mathbf{X}_{i \bmod K}, y_{i \bmod K}) = (y_{i \bmod K} - \varphi(\mathbf{X}_{i \bmod K}))^2 \quad (2)$$

$$\mathscr{L}_2(\mathbf{X}_{i,i+1 \bmod K}) = \left\| u_{i \bmod K}(\mathbf{X}_{i \bmod K}) - u_{i+1 \bmod K}(\mathbf{X}_{i+1 \bmod K})\right\|_F^2 \quad (3)$$

The loss function in Equation (1) is formulated to optimize the parameters $\theta$. It aims to minimize the overall loss by integrating multiple components, including $\mathscr{L}_1$, $\mathscr{L}_2$, and regularization terms.

$\mathscr{L}_1$: This term represents the first component of the loss function, capturing the discrepancy between the predicted outputs and the true labels. Specifically, $y_{i \bmod K}$ denotes the label of organization $O_i$. The form of the objective function $\varphi$ depends on the nature of the sharing task, such as classification or regression, and the chosen local model.

$\mathscr{L}_2$: The term $\mathscr{L}_2$ typically corresponds to a regularization technique, such as L2 regularization, which helps prevent overfitting and promotes model generalization, where $u$ denotes the representation converted from the original data, and $\| \cdot \|_F^2$ refers to the square of the Frobenius norm. It penalizes large parameter values to create a more balanced and robust model.

$\gamma$: This parameter $\gamma$ represents the weight assigned to the $\mathscr{L}_2$ component in the overall loss function. Adjusting $\gamma$ controls the trade-off between fitting the training data and applying regularization.

$\lambda$: The parameter $\lambda$ determines the weight assigned to the regularization terms that penalize the magnitude of the parameters $\theta_{i\bmod K}$ and $\theta_{i+1\bmod K}$. It controls the strength of the regularization, helping manage the model's complexity.

Based on the above illustration, Equation (1) represents a combined loss function designed to optimize the parameters $\theta$. This function integrates the task-specific loss $\mathscr{L}_1$, a regularization term $\mathscr{L}_2$, and a regularization of parameter magnitudes. The components are weighted by $\gamma$ and $\lambda$ to achieve a balance between data fitting and model complexity control.

To ensure the confidentiality and integrity of the local model and global model generation, `MatSwarm` incorporates TEE utilizing Intel SGX. Each organization's cloud server is enabled with Intel SGX and includes $M(M \geq N)$ blockchain nodes within the blockchain network. Each organization can deploy multiple blockchain nodes. To facilitate understanding, we assume that each organization has deployed a single blockchain node, denoted as $BN_i$ ($i \leq M$). The blockchain node $BN_i$ must be deployed on an Intel SGX-enabled cloud server. To train local and global models, each organization requests the creation of two AEs. $AE_1$ is used to load encrypted local datasets and smart contract SC1 for training local models. $AE_2$ is used to load encrypted local model parameters and smart contract SC2 for aggregating local model parameters.

## Working mechanisms

The overall working mechanism of `MatSwarm` includes three main stages: task submission, task execution, and task archive. Videos on the procedures and operations of `MatSwarm` are available as Supplementary Movies 2 and 3.

1) Task Submission: assume that all material organizations willing to conduct joint model training have registered and stored their metadata on the blockchain. Organization $O_1$, as the task issuer among participants, initiates a retrieval transaction request to the local blockchain node with a task information digest. The local blockchain node retrieves the blockchain history to determine whether an archived task related to the task information digest has been generated. If such a task exists, the corresponding retrieval result is returned. If organization $O_1$ does not obtain retrieval results for the archived task, it will retrieve the metadata of organizations from the blockchain. Once the task issuer identifies the organizations to be invited, it will design the sharing task scheme, including the task description, metadata description, and the selection of local models and aggregation methods. The task issuer subsequently initiates a sharing transaction request to organizations with relevant datasets to join the sharing task as participants. The blockchain nodes of participating organizations become active nodes, while those of non-participating organizations remain passive. The active nodes participate in the global model consensus mechanism for the task.

2) Task Execution: to facilitate model aggregation, it is essential to standardize the structure and format of the input datasets among participants. The task issuer should create a virtual dataset and broadcast it to the blockchain network, enabling each participant to align their local datasets with the standardized format. Subsequently, participants can use their standardized datasets to train their local models. The task issuer trains a local model and deploys the code into Smart Contract 1 (SC1) running in its $AE_1$. Other participants can invoke SC1 via the blockchain to train their local models in a similar manner, ensuring uniformity in the local model training code. After each round of local model training, remote attestation is required between the $AE_1$ of each participant to verify the credibility of the remote nodes and the integrity and confidentiality of the local model. Following remote attestation, encrypted local models are shared among organizations to generate the global model.

To ensure the integrity and confidentiality of the aggregation process, each organization's blockchain node performs model aggregation in its $AE_2$. The steps involved are as follows: the task issuer deploys the aggregation algorithm onto the smart contract SC2 running in the $AE_2$. Other participants invoke SC2 from the blockchain, subsequently loading the smart contract and encrypted local model sets submitted by others into their respective $AE_2$. Each participant's $AE_2$ independently aggregates the models to generate a global model. To ensure the credibility of each organization's $AE_2$ and the integrity of the global model, the blockchain network must receive all attestation reports generated by each organization's $AE_2$. Consequently, the blockchain nodes complete remote attestation through a consensus mechanism.

3) Task Archive: during each round of training, organizations obtain the current global model and use it to update their local model until the loss function converges to a specific threshold. However, before a credible global model is ultimately generated, a consensus must be reached among participants. Once a consensus is achieved, the global model is stored on the blockchain to prevent tampering. Therefore, participants must ensure that the final global model is recognized by all participants through a consensus mechanism. The model is then securely stored for future retrieval and use.

**Local model generation.** The initial step in local model training involves loading the encrypted local dataset. To ensure security, Intel SGX's AE only accepts encrypted data. Therefore, before sending the local dataset $DL_i$ to the local $AE_1$, the blockchain node $BN_i$ must encrypt it using a symmetric encryption algorithm such as advanced encryption standard (AES)[42] or Triple Data Encryption Standard[43], represented as $Er(.\,, Kr)$. Symmetric encryption and decryption between $BN_i$ and its $AE_1$ are performed using the key $Kr_i$. This process is denoted as $Er(.\,, Kr_i)$. The key $Er(.\,, Kr)$ is transmitted through a secure channel established by the Diffie-Hellman key exchange protocol[44], which allows two parties to establish a shared secret over an unsecured communication channel, providing a foundation for encrypting further communications. $BN_i$ generates an encrypted local dataset $Er(DL_i, Kr_i)$ and sends it to $AE_1$. Upon receipt, $AE_1$ uses the key $Kr_i$ to decrypt $Er(DL_i, Kr_i)(1 \leq i \leq M)$, obtaining the plaintext $DL_i$ of the local dataset. This process can be represented as $BN_i|Er(DL_i, Kr_i) \rightarrow AE_1|Dr(Er(DL_i, Kr_i), Kr_i)$. The second step involves local model training. Organizations deploy the local model using the smart contract SC1 to train their local datasets in $AE_1$.

For local model training, participants on our platform can select the appropriate machine learning model based on their task requirements, including MLP[45], Lasso[46], RNN[47], and LSTM[48]. As the platform evolves, it will offer a broader range of local training models to meet diverse task requirements. In this paper, we demonstrate the local model training process using an example of predicting perovskite formation energy, employing the MLP neural network for local model training and the stochastic gradient descent algorithm for parameter updating.

As shown in Fig. 7, consider the training between organization $O_1$ and organization $O_2$ as an example. Organization $O_2$ calculates intermediate results and encrypts them with the public key $\mathbf{PK}_{IAS}$ from Intel Authentication Service (IAS). The encrypted intermediate results $[\mathbf{u}_2^t]_{\mathbf{PK}_{IAS}}$ and $[\boldsymbol{\theta}_2^t]_{\mathbf{PK}_{IAS}}$ are sent to organization $O_1$. The organization $O_1$ decrypts the intermediate results using the private key $\mathbf{PR}_{IAS}$ and calculates the local model gradient $\frac{\partial \mathscr{L}_1^t}{\partial \boldsymbol{\theta}_1^t}$ and loss function $\mathscr{L}_1^t$. Similarly, organization $O_1$ calculates a set of intermediate results, encrypts them with the public key $\mathbf{PK}_{IAS}$, and sends the encrypted intermediate results $[\boldsymbol{\theta}_1^t]_{\mathbf{PK}_{IAS}}$ and $[\mathbf{u}_1^t]_{\mathbf{PK}_{IAS}}$ are sent to organization $O_2$, which then calculates the local model gradient $\frac{\partial \mathscr{L}_2^t}{\partial \boldsymbol{\theta}_2^t}$ and loss function $\mathscr{L}_2^t$. Both organizations update their local model parameters $\boldsymbol{\theta}_1^{t+1}$ and $\boldsymbol{\theta}_2^{t+1}$ using the calculated local model gradients. After each organization

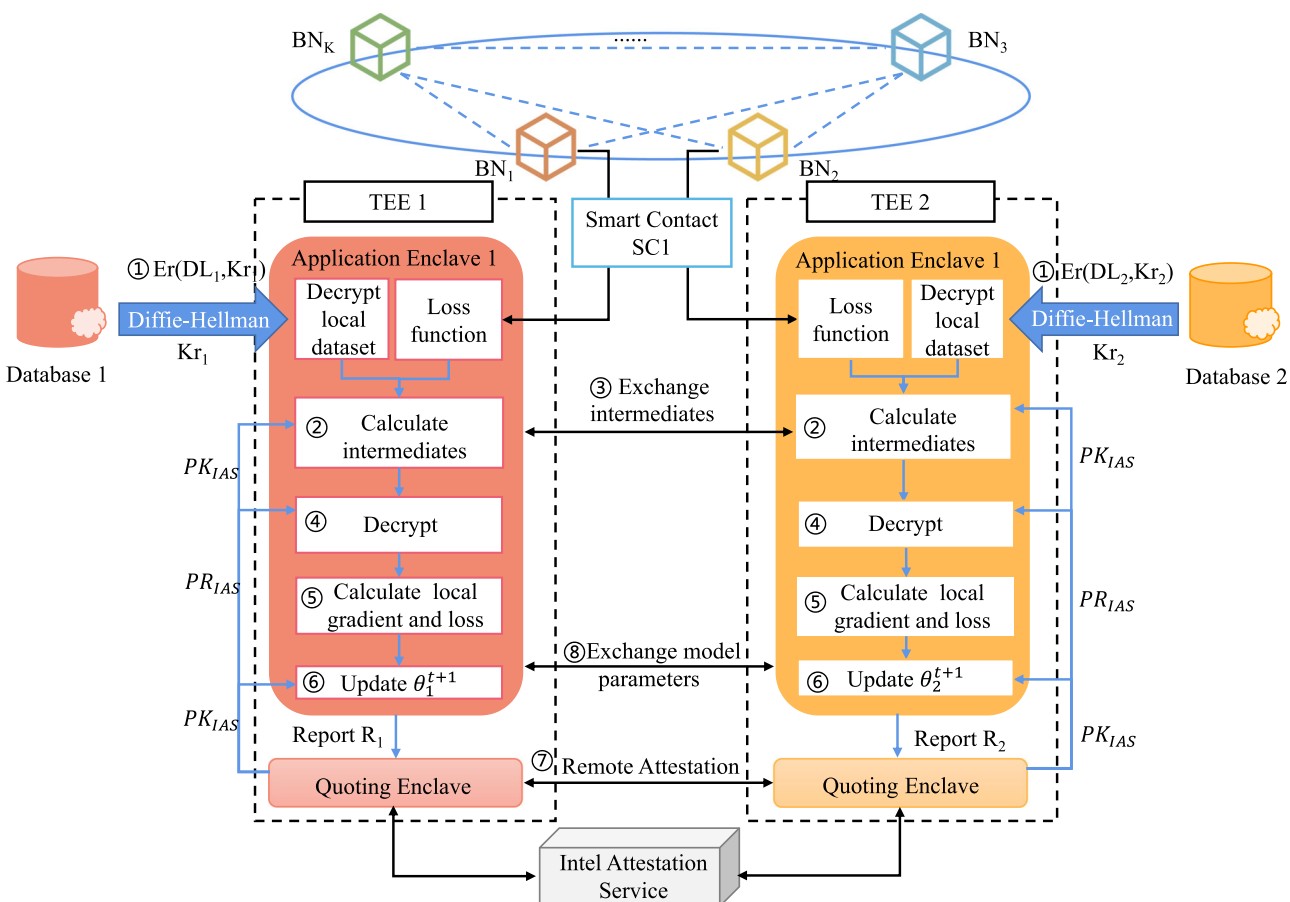

**Fig. 7 | The flowchart of local model generation.** The blue dashed lines represent messages passing between blockchain nodes. The blue solid lines represent message passing within the trusted execution environments (TEEs) of local organizations. The black solid lines represent message passing between trusted execution environments across organizations. Initially, the local blockchain node (BN) invokes the smart contract (SC1) generated by the local model, and then both the smart contract and the encrypted local dataset (DL), using a symmetric key (Kr) established via the Diffie-Hellman protocol, are loaded into the Trusted Execution Environment (TEE). In Application Enclave 1 (AE1), the dataset is decrypted, and intermediate results are calculated. These encrypted results, using a public key (**PK**$_{IAS}$) issued by Intel Attestation Service, are then securely exchanged between organizations through the blockchain network. After exchanging intermediates, (AE1) decrypts the received data and calculates the local model gradient and the loss function. The local model parameters $\theta^{t+1}$ are subsequently updated. Before exchanging model parameters, the Quoting Enclave within the TEE, along with the Intel Attestation Service (IAS), performs remote attestation to verify the integrity and security of the TEEs in all participating organizations.

completes this round of local model training, the blockchain nodes of each organization perform remote certification of all AE$_1$ through a consensus mechanism. Subsequently, organizations encrypt and share the updated local model parameters $\theta_i^{t+1}$ with other participants to aggregate local model parameters for the current round.

**Global model generation.** This section will elaborate on generating global models, covering crucial aspects such as smart contract deployment, model aggregation, remote attestation, and consensus mechanisms. Figure 8 illustrates the process of global model generation.

1) Smart contract deployment: the task issuer $O_1$ deploys the model aggregation algorithm to the blockchain via the local blockchain node BN$_1$ as a smart contract. Each participant can retrieve and invoke the smart contract from the blockchain. The blockchain node BN$_i$ loads the smart contract and encrypted local model parameter set $[[\mathbf{M}_L^t]]_{\mathbf{PK}_{IAS}}$ into AE$_2$ of TEE$_i$. The parameters are then decrypted using **PR**$_{IAS}$ to construct the plaintext of the local model parameter set $\mathbf{M}_L^t = \left(\theta_1^t, \theta_2^t, \cdots, \theta_K^t\right)$. The calculation of global model parameters occurs in AE$_2$ of TEE$_i$ to ensure the confidentiality of sensitive parameters. Smart contracts facilitate the transfer of global model parameters.

2) Model aggregation: the global model parameters $\mathbf{M}_G^{t+1}$ calculated in each participant's AE$_2$ are given by:

$$\mathbf{M}_G^{t+1} = \sum_{i=1}^{K} \frac{|DL_i|}{n} \mathbf{M}_{L_i}^t, \; n = \sum_{i=1}^{K} |DL_i| \qquad (4)$$

where $\mathbf{M}_G^{t+1}$ represents the global model updated in the $t+1$ round, $K$ denotes the number of participants, $|DL_i|$ represents the number of samples used by the $i$-th participant to train the local model, and $n$ is the total number of samples used to train all local models. $\mathbf{M}_{L_i}^t$ is the local model parameters set updated by the $i$-th participant in the $t$ rounds.

Notably, parameter aggregation is illustrated using the Mean method[35] in `MatSwarm`, which is the most widely used approach. However, various aggregation methods are available, such as MultiKrum[49], CenteredClipping[50], GeoMed[51], and Median[52], among others. On our platform, participants can choose different model aggregation methods based on task requirements and robustness needs.

3) Remote attestation: during the global model generation process, remote attestation is used to verify the integrity of the global model generated by AE$_2$. In this method, the blockchain node BN$_i$ facilitates interaction between the AE$_2$ of TEE$_i$ and

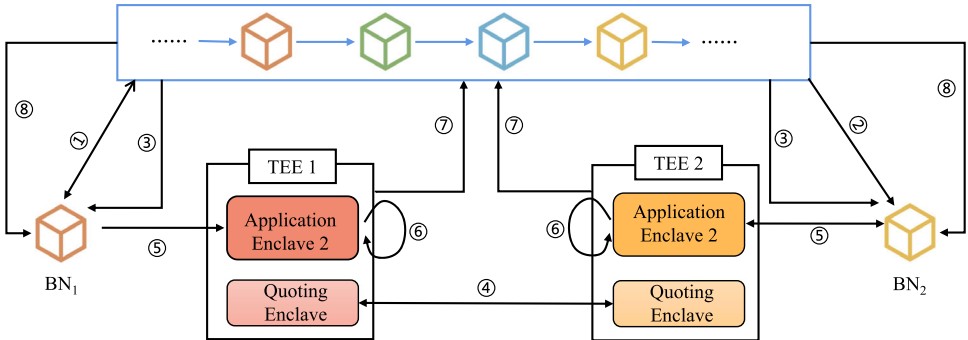

**Fig. 8 | The flow diagram of the local model parameters aggregation process.**
Initially, the task issuer's blockchain node (BN₁) deploys the model aggregation algorithm as a smart contract (SC2), and other blockchain nodes invoke it. All participants' blockchain nodes extract the encrypted local model parameters. Before sending SC2 and local model parameters into Application Enclave 2 (AE2) of the Trusted Execution Environment (TEE), remote attestation is performed to verify the integrity of the global model computation. The global model is then computed within AE2, and the resulting global model parameters are sent to the Quoting Enclave for generating Quotes (**QG**). These Quotes are used in the consensus process to finalize and store the global model on the blockchain.

the blockchain network, serving as both an aggregator and verifier in the attestation process. $TEE_i$ generates a REPORT structure information $\mathbf{R}_i$ containing the current enclave identity information $\mathbf{M}_i$, and other metadata through the EREPORT function, and signs $\mathbf{R}_i$ to produce a Message Authentication Code (MAC) tag $MAC_i$. $AE_2$ sends $\mathbf{R}_i$ and MAC tags to the Quoting Enclave in $TEE_i$ for mutual attestation. The Quoting Enclave calls the EGETKEY command to decrypt the $MAC_i$ and verifies the decrypted information against $\mathbf{R}_i$. After successful mutual attestation within $TEE_i$, the Quoting Enclave uses the private key ($\mathbf{AKr}_i$) of the attestation key generated by the Intel Provisioning Service's Provisioning Seal Key, to sign $\mathbf{R}_i$ and create a Quote $\mathbf{QG}_i = Sign(\mathbf{R}_i, \mathbf{AKr}_i)$. Only Quoting Enclave can access the key used for attestation in the Intel Provisioning Service to verify the credibility of $TEE_i$. The $\mathbf{QG}_i$ is then sent through the blockchain network to the blockchain nodes of other participants for verification.

Once $BN_i$ receives K-1 Quotes, it will verify each Quote using the public key $\mathbf{AKp}_i$ generated by the Intel Provisioning Service. The verification is completed utilizing the function $verify(Sign(\mathbf{QG}_i, \mathbf{AKr}_i), \mathbf{AKp}_i)$. Once the Quote is validated, $BN_i$ will extract the global model $\mathbf{M}_G^{t+1} = \left(\mathbf{M}_{G_1}^{t+1}, \mathbf{M}_{G_2}^{t+1}, \ldots, \mathbf{M}_{G_K}^{t+1}\right)$ from $\mathbf{QG} = (\mathbf{QG}_1, \mathbf{QG}_2, \ldots, \mathbf{QG}_K)$ for subsequent consensus.

4) Global Model Consensus: at this stage, the consensus mechanism is used to determine the global model accepted by the participants. We use the PBFT[53] consensus, which can tolerate $f$ Byzantine fault nodes. We assume that three organizations are participating in the shared task. The blockchain node $BN_1$ is a blockchain node of the task issuer $O_1$ acting as the primary node; $BN_2$ and $BN_3$ are blockchain nodes of the other two participants $O_2$ and $O_3$ participating in the consensus mechanism as active nodes; $BN_j$ $(j \in M)$ denotes the blockchain node of organizations that are not participating in the sharing task, referred to as passive nodes. The consensus mechanism consists of five steps: request, pre-prepare, prepare, commit, and reply.

Request phase: the task issuer $O_1$ initiates a global model consensus request to the deployed blockchain node $BN_1$.

Pre-prepare stage: $BN_1$ calculates $Hash(\mathbf{M}_{G_1}^{t+1}, \mathbf{M}_{G_2}^{t+1}, \ldots, \mathbf{M}_{G_K}^{t+1})$ and broadcasts $Hash(\mathbf{M}_{G_1}^{t+1})$ to $BN_2$ and $BN_3$ if the Hash of all global models is equal.

Prepare stage: after receiving the $Hash(\mathbf{M}_{G_1}^{t+1})$ sent by $BN_1$, $BN_2$, and $BN_3$, calculate the Hash value of the global model $\mathbf{M}_{G_i}^{t+1}(1 \leq i \leq K)$ sent by each organization. If all hash values are equal to $Hash(\mathbf{M}_{G_1}^{t+1})$, $BN_2$ and $BN_3$ broadcast $Hash(\mathbf{M}_{G_2}^{t+1})$ and $Hash(\mathbf{M}_{G_3}^{t+1})$ to the other two participants, respectively.

Commit stage: after receiving the calculation results from the other nodes, all participants verify whether a consistent global model has been agreed upon by all. If consensus is achieved, they broadcast confirmation messages to the other participants.

Reply stage: the consensus request is considered complete when each participant receives confirmation messages from at least two-thirds of the nodes. A Reply message is then constructed and sent to $O_1$. Once $O_1$ receives confirmation messages from more than two-thirds of the nodes, it finalizes the global model and broadcasts its hash to all active and passive nodes for storage.

The processes of local and global model generation are repeated round and round until the model converges to a threshold. In the end, the final global model is stored on the blockchain, ensuring tamper resistance while facilitating efficient retrieval by others.

### Reporting summary
Further information on research design is available in the Nature Portfolio Reporting Summary linked to this article.

## Data availability
All datasets used are publicly available at https://github.com/SICC-Group/MatSwarm.git and Zenodo[54]. All data supporting the findings described in this manuscript are available in the article and in the Supplementary Information and from the corresponding author upon request. Source data are provided with this paper.

## Code availability
The codes are available in open source at https://github.com/SICC-Group/MatSwarm.git and Zenodo[54].

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

## Acknowledgements

This work is supported in part by the National Key Research and Development Program of China under Grant 2021YFB3702403, and in part by the National Natural Science Foundation of China under Grant 62101029. R.W. has been supported by the China Scholarship Council Award under Grant 202306460078. C.X. has been supported in part by the China Scholarship Council Award under Grant 202006465043.

## Author contributions

R.W. and C.X. conceived this project. C.X. and X.Z. funded and supervised the research. R.W. and F.Y. implemented the algorithm, performed the experiments, and prepared the plots. Y.T., S.T., H.Z., and W.D. implemented the open-source prototype. R.W. and C.X. analyzed the results and drafted the main text. C.X., S.Z. and X.Z. revised the manuscript. All authors commented on the manuscript.

## Competing interests

The authors declare no competing interests.
