## [Peer Review File · Nature Communications]

MatSwarm: Trusted Swarm Transfer Learning Driven Materials Computation for Secure Big Data SharingReviewers' Comments:

Reviewer #1 (Remarks to the Author):

The manuscript by Wang et al describes MatFed a framework for federated transfer learning, i.e., for collaborative machine-learning model training without exposing sensitive information on the (training+test) data. A described example concerns perovskite materials data, otherwise it is unclear in which sense MatFed is tailored for materials data, as claimed. It is clear that the platform is introduced in the context of the National Materials Genome Engineering (MGE) project, but there is nothing in the description that clarifies in which sense MatFed specifically addresses Materials Science data.

Overall, I recognize the usefulness of the described platform, and possibly also its revolutionary impact, at least in some context of model training and utilization, but overall I do not see the intrinsic scientific significance of the project. All the blockchain-related aspects are certainly not novel as essentially recognized by the authors. For instance, I surmise that MatFed is heavily based on Hyperledger Fabric.

There are several levels at which I think the manuscript and the overall project are not scientifically relevant:

- the knowledge of the blockchain technology is assumed known by the reader. The manuscript is rather detailed with respect to the several actions available to the MatFed users, but the whole text is full of jargon (blockchain themselves, smart contract, Diffie-Hellman security exchange protocol, ...).
- can a reader replicate MatFed after reading the manuscript? I doubt so. I read descriptions of schemas but I see nothing that can be checked. MatFed itself seems to have no internet presence and indeed no reference URL is given in the manuscript. The software availability section claims the software can be asked to the corresponding author, but this is a completely noncommittal statement. Especially the "reasonable request" wording leave the authors completely . So, if one wants to publish a scientific paper, the presented results should at least in principle be reproducible and/or checkable. The community is striving for FAIR scientific data and software, where is the FAIRness, here?
- how can one ever analyze the model and its generalization performance without knowing the data? E.g., one should be able to perform some form of a posteriori explainability analysis, which eventually identifies data responsible for a given prediction. Similarly, one may want to perform outlier detection to ascertain whether out-of-distribution data are simply "wrong" in some sense or signaling some interesting physics. Without these and similar tools, there is a specific decision in completely black-boxing predictive models, which is at odds with scientific research.
- is the (encrypted) access to local gradients (section 4.2) just an example? Not all ML model classes use gradients wrt parameters to iteratively train the models, isn't the the platform essentially limiting the type of model classes one can apply?
- is the global model (Eq. 6) the weighted average of the local models? This look like an ensemble model, which has its merits, but why is this the only option?

In summary, it is possible MatFed will significantly impact the way materials science research is performed, but the current manuscript lacks several aspect of a good scientific publication, as briefly described above. I therefore cannot recommend its publication.

Reviewer #2 (Remarks to the Author):

The paper introduces an architecture for federated learning in material science that uses block chain to avoid the central server federating models.

The paper shows the main architecture together with the functions to finally show in a real use case how the proposed framework behaves.

Despite the work is interesting and in material sciences the use of federated learnign using block chain has not been used, the main issue with the proposed approach is that it does not mention swarm learning. In fact all the references in the related work are prior to 2022 when swarm learnign started to be used.

As a consequence, the framework proposed is not so innovative as the authors claims.

Besides the paper does not include a nice discussion section in which one can see the pros and cons of the presented approach. The paper presents the resutls on a datasets but is is difficult to reproduce the research.

On the other hand, the framework that has been presented by the author in this case for federated learning is very similar to the one presented in the paper entitled: S-MBDA: A Blockchain-Based Architecture for Secure Storage and Sharing of Material Big-Data by the same authors.

On the other hand, the lack of review of related works in the last year in such a rapid evolving field together with the lack of conclusions and discussion on the results presented makes the reserch presented not mature enough for publication.

Dear Reviewers:

The authors appreciate your valuable comments. We found them very helpful in revising the paper. We have carefully examined these comments and made appropriate changes to the revised paper. Here we have outlined the details of our responses to the specific comments from the reviewers.

1 Responses to Reviewer 1

Comment 1.1

The manuscript by Wang et al describes MatFed a framework for federated transfer learning, i.e., for collaborative machine-learning model training without exposing sensitive information on the (training+test) data. A described example concerns perovskite materials data, otherwise it is unclear in which sense MatFed is tailored for materials data, as claimed. It is clear that the platform is introduced in the context of the National Materials Genome Engineering (MGE) project, but there is nothing in the description that clarifies in which sense MatFed specifically addresses Materials Science data.

Response:

We would like to express our gratitude for your valuable feedback and constructive comments on our manuscript. Your insights are highly appreciated, and we acknowledge that we indeed overlooked explaining how our proposed *MatSwarm*¹ framework addresses the prominent challenges encountered in the collaborative computing process of material data. We failed to clearly articulate the characteristics of material data and the key scientific issues it faces. Therefore, we have revised the introduction to include the prominent challenges faced in sharing and utilizing material data, and highlight the main modifications in red. We detail how our secure collaborative computing framework, designed for big data in materials, addresses these challenges. Moreover, our solution enhances model accuracy and generalization capabilities without compromising the security of original data, serving as a reference for collaborative modeling among various organizations in the materials field.

To clarify, *MatSwarm* incorporates several features designed specifically for materials science data:

- *Swarm Learning Enhancements*: Within the *MatSwarm* framework, we propose a swarm transfer learning method to enhance swarm learning techniques, address-

¹Based on the feedback from another reviewer, noting that blockchain-based federated learning is often defined as swarm learning by relevant researchers, we have accordingly redefined the framework proposed in this paper as *MatSwarm* (formerly *MatFed*).

ing challenges unique to materials science, such as non-independent and identically distributed (non-i.i.d.) material data and the need for fine-tuning on domain-specific datasets. The core of our method lies in identifying invariances between resource-rich source domains and resource-scarce target domains. This facilitates the learning of common representation spaces and enables knowledge transfer across domains, ensuring robust generalization across heterogeneous material data.

- *Integration with Materials Databases:* The *MatSwarm* framework is designed to seamlessly integrate with established materials databases, allowing for efficient data sharing and model training across institutions involved in the NMDMS platform. This integration ensures that the collaborative computing process is grounded in comprehensive and high-quality datasets specific to materials science.
- *Domain-Specific Feature Engineering:* The *MatSwarm* framework integrates domain-specific feature engineering techniques that extract meaningful features from materials datasets. For instance, features related to atomic radii, bonding characteristics, and electronic structures are systematically generated to enhance model performance on materials-related tasks. These feature preprocessing steps ensure that the data fed into the machine learning models retains the essential characteristics required for accurate predictions in materials science.

For non-i.i.d. material data and issues such as insufficient data samples and features, existing swarm learning solutions fail to achieve a high-precision and highly generalized training model. Through *MatSwarm*, each organization can fully learn the sample data characteristics of other organizations. Even with non-i.i.d. training datasets and limited samples and features, *MatSwarm* can achieve a high-precision and well-generalized training model.

In summary, the key contributions of this work are outlined as follows:

1. ***Innovative Big-Data Security Collaborative Computing Framework for the Material Science Domain:*** We propose *MatSwarm*, which is a decentralized collaborative computing framework, tailored for the materials field, addressing the collaborative training needs for sensitive datasets. To the best of our knowledge, the application of the *MatSwarm* framework in the materials field is unprecedented. We utilized real datasets from the National Material Data Management and Services (NMDMS) platform to verify its applicability and performance advantages. Our empirical results provide substantial insights for researchers aiming to construct similar frameworks, significantly enriching the discourse in this domain.
2. ***Swarm Transfer Learning Adapted for the Heterogeneity of Material Data:*** Recognizing the non-i.i.d. of material data from various sources,

we advocate an innovative swarm transfer learning method based on swarm learning. This method allows each material organization to obtain a model with higher accuracy and better generalization ability while preserving its unique features. Even with non-i.i.d. training datasets and limited samples and features, our proposed method achieves high-precision and well-generalized training models.

- 3. *Swarm Transfer Learning Combined with TEE to Enhance Confidentiality:*** *To safeguard the confidentiality of model parameters and effectively mitigate risks associated with model tampering, we incorporate the Trusted Execution Environment based on Intel Software Guard Extensions (SGX). Each participating material research organization is equipped with at least one blockchain node powered by an SGX-enabled processor, responsible for executing model training tasks. During the model aggregation phase, the confidentiality of the aggregated model is verifiable by each node. This setup ensures confidentiality, integrity, and prevents inaccurate model aggregations.*

We ensure that these details are explicitly highlighted in the revised version of our manuscript to better convey how *MatSwarm* is tailored to the unique requirements of material data. Detailed information could be referred to the Introduction of the main text.

Comment 1.2

Overall, I recognize the usefulness of the described platform, and possibly also its revolutionary impact, at least in some context of model training and utilization, but overall I do not see the intrinsic scientific significance of the project. All the blockchain-related aspects are certainly not novel as essentially recognized by the authors. For instance, I surmise that MatFed is heavily based on Hyperledger Fabric.

Response:

Thank you for your insightful comments on our manuscript. We greatly appreciate your recognition of the potential impact of our platform in the context of model training and utilization. In response to your concern regarding the intrinsic scientific significance of our project, we would like to provide further clarification and additional context that underscore the novelty and scientific contributions of our work, especially beyond the blockchain aspects.

While blockchain technology itself is not novel and primarily provides a decentralized secure network architecture for our platform, the innovation in our work lies in the unique integration of a trusted execution environment and swarm transfer learning

specifically tailored for the materials science domain. This integration addresses critical challenges in data security and data sharing that are not effectively solved by existing methods. Our framework, *MatSwarm*, enables collaborative model training across various institutions without exposing sensitive data, thereby advancing the field of material big data secure sharing in a novel manner. *MatSwarm* has been established and put into use in China, significantly accelerating research and development processes in the materials industry.

1. *Limited Access to Diverse Datasets*: Our platform significantly mitigates the challenge of accessing diverse and extensive datasets in materials science. By enabling collaborative model training, *MatSwarm* allows for the utilization of distributed data sources without requiring data transfer, thus expanding the available data pool while maintaining confidentiality. **To the best of our knowledge, our proposed collaborative computing framework is the first of its kind in the field of materials science.**
2. *Data Security in Collaborative Environments*: We introduce a novel *MatSwarm* framework to secure collaborative computing by implementing a swarm transfer learning method that operates within a Trusted Execution Environment (TEE). This setup ensures that data and model integrity are maintained even in multi-party collaborative scenarios, which is a significant advancement over traditional swarm learning methods.
3. *Enhancing Model Generalization Across Diverse Data Sources*: The federated nature of *MatSwarm* allows it to effectively handle heterogeneous data sources, improving the generalization of models across different materials and conditions. This capability is critical for the discovery and optimization of new materials.
4. *Empirical Validation and Real-world Applicability*: Existing federated learning solutions have predominantly been theoretically validated using publicly available datasets and are often applied to address classification problems. There is a notable lack of empirical validation in real-world applications, raising questions about the practical applicability and feasibility of these solutions. Our empirical results demonstrate the effectiveness of *MatSwarm* in real-world scenarios, as deployed on the NMDMS platform in China, showing notable improvements in model prediction accuracy and data utilization. The application of such a collaborative computing method in the materials field is unprecedented and provides substantial insights for researchers aiming to construct similar frameworks.

In summary, while individual components such as blockchain might not be novel, the scientific significance of our work arises from the innovative combination of these

technologies to address specific, previously unsolved problems in materials science research. This approach not only enhances the security and efficiency of data sharing and model training but also contributes to the field's body of knowledge by enabling new capabilities in predictive modeling and materials discovery. This is an unprecedented work in the field of material big data and has great significance for actual industrial production and material science research.

Comment 1.3

the knowledge of the blockchain technology is assumed known by the reader. The manuscript is rather detailed with respect to the several actions available to the MatFed users, but the whole text is full of jargon (blockchain themselves, smart contract, Diffie-Hellman security exchange protocol, ...).

Response:

Thank you for your constructive feedback on our manuscript. We acknowledge your concerns regarding the presumed knowledge of blockchain technology among our readers and the use of technical jargon throughout the text. In response to your comments, we have taken several steps to make the manuscript more accessible and informative for all readers, including those who may not have a background in blockchain technology.

1. *Clarification and Definitions of Technical Terms:* We have added clear definitions and explanations for key technical terms such as "smart contract" and "Diffie-Hellman security exchange protocol" at their first occurrence within the text. These explanations are designed to provide a brief overview of the concepts for unfamiliar readers. All added definitions and significant changes to the text have been highlighted in red to ensure they are easily noticeable by reviewers.
2. *Referencing and Further Reading:* To aid in-depth understanding, we have incorporated relevant references next to technical terms. These references are carefully selected to guide readers towards comprehensive resources for further reading on blockchain technologies and their applications. For instance, for the Diffie-Hellman key exchange protocol, we have referenced a well-regarded source that discusses its implementation and significance in secure communications.

The supplementary is as follows:

The key $E_r(., Kr)$ is transmitted through a secure channel established by the Diffie-Hellman key exchange protocol [51], which enables two parties to establish a shared secret over an unsecured communication channel, providing a foundation for encrypting further communications.

This approach ensures the integrity of model aggregation, and all organizations automatically execute the same model aggregation code through smart contracts [23], which automates the enforcement and management of agreed-upon processes and conditions, ensuring consistent execution and eliminating discrepancies, while also enhancing security and reducing reliance on third-party intermediaries.

3. *Technical Review and Simplification:* Beyond just blockchain-related terms, we have reviewed the entire manuscript for any potential technical jargon and provided simplified explanations or relevant references where necessary. This effort extends to terms related to swarm learning and other specialized fields mentioned in our study.

These modifications aim to demystify the technical jargon discussed in our paper and make the content accessible and engaging for a diverse audience. We believe that these changes address the reviewer's concerns effectively and enhance the manuscript's readability.

Comment 1.4

can a reader replicate MatFed after reading the manuscript? I doubt so. I read descriptions of schemas but I see nothing that can be checked. MatFed itself seems to have no internet presence and indeed no reference URL is given in the manuscript. The software availability section claims the software can be asked to the corresponding author, but this is a completely noncommittal statement. Especially the "reasonable request" wording leave the authors completely . So, if one wants to publish a scientific paper, the presented results should at least in principle be reproducible and/or checkable. The community is striving for FAIR scientific data and software, where is the FAIRness, here?

Response:

Thank you for your constructive feedback on our manuscript. We appreciate your concerns regarding the replicability of the *MatSwarm* framework as described in our document.

Replicability and Availability of *MatSwarm*: Regarding the replicability of our framework, we understand the importance of making scientific research accessible and reproducible. As mentioned by the reviewers, the original intention of the NMDMS platform construction is to accelerate material sharing and improve the flow and sharing efficiency of data in the field. The *MatSwarm* framework proposed in this paper aims to better serve the material community and enhance data sharing. However, the *MatSwarm* framework was developed as part of the Material Genome Engineering (MGE) project,

which involves sensitive data and is subject to national security and confidentiality requirements. Consequently, we are unable to provide direct access to the complete source code of the *MatSwarm* framework.

To address this and still support the scientific community, we have developed a demonstration system that integrates the core functionalities of the *MatSwarm* framework, which can be accessed at <https://github.com/SICC-Group/MatSwarm.git>. Detailed operational procedures can be found in the supplementary videos. This demo system is designed to allow researchers to replicate key aspects of our work and explore the capabilities of the framework under conditions that adhere to the security protocols required by the MGE project. For data access, *MatSwarm* is built upon the NMDMS platform, which is open to the community, and any registered user can access the open data on the platform (currently more than 14 million pieces). Sensitive or non-public data is not directly available, and the *MatSwarm* framework's original intention is to facilitate the use of such data under secure conditions.

Commitment to FAIR Principles: In line with our commitment to the FAIR principles—making our data Findable, Accessible, Interoperable, and Reusable—we are providing detailed documentation and step-by-step guides within the demo system. These resources are crafted to ensure researchers can effectively use the system and independently verify our claims. The supplementary information includes:

1. *Code Availability:* We make the complete source code of *MatSwarm* available on a public repository, ensuring it is accessible to all researchers. The repository includes detailed documentation on setting up and running the code.
2. *Detailed Replication Steps:* The public repository provided in our manuscript now includes a supplementary section detailing step-by-step instructions on how to replicate the experiments and use the *MatSwarm* framework. This will ensure that researchers can independently verify our claims.
3. *Demonstration Video:* We have provided a supplementary video demonstration of the *MatSwarm* framework in action, showcasing its application in predicting the formation energy of perovskite materials. This visual aid will help researchers better understand the framework's functionality and integration.

We believe these adjustments effectively address the reviewers' concerns and enhance the manuscript's research and scientific value by ensuring transparency and reproducibility within the constraints imposed by security requirements.

Comment 1.5

how can one ever analyze the model and its generalization performance without knowing the data? E.g., one should be able to perform some form of a posteriori explainability analysis, which eventually identifies data responsible for a given prediction. Similarly, one may want to perform outlier detection to ascertain whether out-of-distribution data are simply "wrong" in some sense or signaling some interesting physics. Without these and similar tools, there is a specific decision in completely black-boxing predictive models, which is at odds with scientific research.

Response:

Thank you for your insightful comments regarding the transparency and analysis of the model and its generalization performance in our manuscript. Your concern about the necessity of understanding the data to analyze and validate predictive models effectively is indeed pivotal for advancing scientific research. In response to this comment, we offer the following explanations:

"Data silos"-an intractable problem in real-world applications of materials science: In practical applications, organizations are often reluctant to share sensitive or commercially valuable raw data with others, even if it compromises model training and generalization analysis. This reluctance to share data is an objective reality that is difficult to change. Consequently, all model and generalization performance must rely on this premise. The *MatSwarm* framework addresses this challenge by enabling collaborative learning without disclosing the original data, while also providing a certain degree of generalization and post-explainability capabilities. Each participating organization on the platform has the flexibility to utilize their local, sensitive data, which may not be publicly available due to confidentiality agreements. These organizations can engage in collaborative training using datasets stored on the platform that are not open to the public. This ensures the integrity and confidentiality of sensitive data while still leveraging the benefits of swarm learning. Additionally, *MatSwarm* provides tools for model analysis without disclosing the original data, as detailed in the response below and in Section 5.2 of the supplementary document.

Experimental Verification of *MatSwarm*'s Generalization Performance: In our experiments, we aim to verify that *MatSwarm* achieves high accuracy and good generalization capability on non-i.i.d material data. Each organization's labeled data is non-i.i.d, as shown in Fig. 5. We use a test set (a new dataset) to evaluate *MatSwarm*'s generalization ability. As depicted in the figure, the label distribution of the test set is inconsistent with the label distributions of the training sets from the three organizations. If the MSE of the test set is low, it indicates that our model has good generalization ability for new datasets. All data and the open-sourced demo system that supports the findings

Fig. 5 Distribution of data set labels for the three participants: (a) non-i.i.d. for training, (b) i.i.d. for training, and (c) test data set.

Fig. 8 Comparison of the influence of different data label distributions on the accuracy of *MatSwarm*: (a) i.i.d.; (b) non-i.i.d..

of this study are available and can be obtained from the following public repository: <https://github.com/SICC-Group/MatSwarm.git>.

As shown in Fig. 8, the experimental results demonstrate that the global model of *MatSwarm* achieves high accuracy and significantly improves the accuracy of the local models of each organization to varying degrees. The prediction MSE for Organization 1 decreased from 1.029 to 0.209, and for Organization 3, it decreased from 1.615 to 0.584, with the global model achieving an accuracy as low as 0.090. Despite Organization 2 having a similar label distribution to the test set and thus showing good prediction accuracy, its local model prediction accuracy also improved slightly after training with *MatSwarm*. This demonstrates that *MatSwarm* has strong generalization capabilities for non-i.i.d. material data.

Analytical Tools and Model Explainability: Addressing your point on model explainability and outlier detection, our system is designed to incorporate advanced analytical tools that allow for a posteriori explainability analysis, even without access to the original data. For example, we use SHAP values to analyze the feature importance

for the perovskite formation energy prediction model. This method enables researchers to identify the data responsible for specific predictions and to ascertain whether out-of-distribution data points are erroneous or indicative of novel scientific phenomena. The integration of this method into our framework aims to mitigate the issue of black-boxing predictive models, fostering a more transparent and scientifically rigorous approach. **For details about the SHAP values, please refer to Section 5.2 of the supplementary document.**

We appreciate your feedback as it has highlighted critical aspects of our project that we are committed to improving. We believe that these clarifications and the features of our demonstration system adequately address your concerns, aligning with the principles of open scientific inquiry and robust model validation.

Comment 1.6

is the (encrypted) access to local gradients (section 4.2) just an example? Not all ML model classes use gradients wrt parameters to iteratively train the models, isn't the the platform essentially limiting the type of model classes one can apply?

Response:

Thank you for your insightful question regarding the adaptability of the *MatSwarm* framework to machine learning models that do not utilize gradients with respect to parameters for iterative training.

The *MatSwarm* framework, as described in Section 4.2 of our paper, primarily employs a swarm transfer learning approach optimized using gradient-based methods, a common strategy for many machine learning models in the materials science domain. This framework has been effectively utilized in the National Materials Genome Engineering (MGE) project, demonstrating promising results in secure big data sharing and collaborative model training without exposing sensitive information.

However, we acknowledge that not all machine learning models rely on gradients for training. For instance, models like Recurrent Neural Networks (RNNs), Long Short-Term Memory (LSTM) networks, and other neural networks frequently used in materials science share training parameters using different strategies. The *MatSwarm* framework can support such models through the principles of swarm learning without directly relying on gradient-based updates. The core of the framework is its ability to securely aggregate model updates, which do not necessarily have to be gradient updates. In our actual deployment platform, a variety of basic models are available for users to choose from, including both gradient-based and parameter-based models, with extensible and customizable types.

In conclusion, the design of our *MatSwarm* framework is not inherently restrictive. **We provide demonstration system videos and other supplementary documents.**

For different training tasks, users can select the appropriate local model and aggregation method according to their specific needs. As the demands of applications grow, we plan to integrate more machine learning models and aggregation methods to fulfill an increasing variety of materials data collaboration needs, ensuring the framework’s applicability and utility for a broad spectrum of collaborative computation tasks in materials science and beyond.

Comment 1.7

is the global model (Eq. 6) the weighted average of the local models? This looks like an ensemble model, which has its merits, but why is this the only option?

Response:

Thank you for your observations and questions regarding the global model formation as described in our manuscript, specifically concerning Equation 6.

You correctly noted that the representation provided in the manuscript suggested that the weighted average of the local models, resembling an ensemble model, was the primary method for aggregating these models. However, this was merely an illustrative example, and we appreciate your pointing out the potential for misunderstanding this as the sole approach available within our framework.

To clarify, our platform supports a variety of aggregation methods, with average aggregation being just one of the options available in our demonstration system. We recognize the importance of providing flexibility in choosing the aggregation method that best fits the specific requirements of the task and the characteristics of the data being analyzed.

In response to your feedback, we have revised the Section 4.3 of the manuscript to clearly articulate that multiple aggregation methods are supported and that the choice of method can be tailored according to the user’s needs. This revision aims to ensure that readers do not misconstrue the average aggregation as the only option. The revised content is as follows:

Notably, parameter aggregation is illustrated using the Mean method [35], which is the most widely used approach. However, there are various aggregation methods available, such as MultiKrum [8], CenteredClipping [31], GeoMed [11], Median [78], among others. On our platform, participants can choose different model aggregation methods based on task requirements and robustness needs.

In addition, we provide a demonstration video of the platform and other supplementary documents, from which you can see that both the local model and the

aggregation method can be chosen by the user. As the platform continues to iterate, we will support the use of more local models and aggregation methods.

We believe these changes enhance the manuscript by more accurately reflecting the capabilities of our *MatSwarm* framework and its applicability to diverse computational tasks in materials science.

2 Responses to Reviewer 2

Comment 2.1

Despite the work is interesting and in material sciences the use of federated learning using block chain has not been used, the main issue with the proposed approach is that it does not mention swarm learning. In fact all the references in the related work are prior to 2022 when swarm learning started to be used. As a consequence, the framework proposed is not so innovative as the authors claims.

Response:

We appreciate your thorough review and insightful comments, particularly regarding swarm learning. Upon further investigation, we found that some researchers define distributed machine learning based on blockchain technology as swarm learning. Taking this into consideration, we have updated the details of our proposed *MatSwarm* framework in the manuscript. Additionally, we have supplemented the related work section with relevant research on swarm learning. The updated content is as follows:

***Blockchain enabled Federated Learning.** In recent years, swarm learning has been proposed by integrating distributed machine learning and blockchain to address the security of raw data during data sharing. The combination of blockchain and FL is one of the typical solutions [52,87,59]. These approaches maintain control over shared raw data and improve the anti-attack capabilities of models during the FL process. For instance, Pokhrel et al. [53] proposed a blockchain-based autonomous federated learning design for privacy awareness and efficient vehicle network communication, although the high computation cost increases overall delay time. Warnat et al. [67] used swarm learning to develop disease classifiers using distributed data, focusing on four use cases of heterogeneous diseases (COVID-19, tuberculosis, leukemia, and lung pathologies). Saldanha et al. [58] have demonstrated the successful use of swarm learning on large, multicentric datasets of gigapixel histopathology images from over 5,000 patients. However, the models in this scheme are stored in blockchain nodes and are not fully encrypted, making them vulnerable to tampering attacks. Although combining blockchain with*

FL has significant potential, the issue of parameter leakage in FL models, which can expose raw data, must be urgently addressed.

While we acknowledge that blockchain-based federated learning frameworks are not novel, our primary contributions lie in two areas: firstly, we introduce a collaborative computing framework and methodology in the materials science domain for the first time; secondly, we have optimized and improved upon the existing swarm learning framework to address the unique challenges posed by heterogeneous materials data in collaborative computing. These challenges include:

Sensitive Protection: In materials science research, data often contain sensitive business secrets or valuable intellectual property information. During the model aggregation and dissemination phases, the model parameters are susceptible to data poisoning attacks, leading to the inversion and leakage of the original data. Therefore, ensuring that data are not disclosed during processing is our primary task.

Handling Data Heterogeneity: Due to differences in sampling equipment and methods, the data labels for each organization are essentially non-independent and identically distributed (non-i.i.d.). A learning method capable of effectively handling this heterogeneity is needed to achieve efficient cross-node model training while keeping the data localized.

Scalability: Considering the varying factors such as training tasks, dataset sizes, feature parameters, and the number of participants, each participant hopes to obtain a highly accurate and well-generalized training model, even with limited local samples and features. Furthermore, each participating party hopes that the platform supports various local models and aggregation methods to cover a wider range of task requirements.

To address these challenges, we propose the *MatSwarm* framework. To clearly highlight these contributions to the readers, we have revised the main contributions in the introduction section. The updated main contributions are as follows:

1. ***Innovative Big-Data Security Collaborative Computing Framework for the Material Science Domain:*** *We propose MatSwarm, which is a decentralized collaborative computing framework, tailored for the materials field, addressing the collaborative training needs for sensitive datasets. To the best of our knowledge, the application of the MatSwarm framework in the materials field is unprecedented. We utilized real datasets from the National Material Data Management and Services (NMDMS)*

platform to verify its applicability and performance advantages. Our empirical results provide substantial insights for researchers aiming to construct similar frameworks, significantly enriching the discourse in this domain.

2. **Swarm Transfer Learning Adapted for the Heterogeneity of Material Data:** Recognizing the non-i.i.d. of material data from various sources, we advocate an innovative swarm transfer learning method based on swarm learning. This method allows each material organization to obtain a model with higher accuracy and better generalization ability while preserving its unique features. Even with non-i.i.d. training datasets and limited samples and features, our proposed method achieves high-precision and well-generalized training models.
3. **Swarm Transfer Learning Combined with TEE to Enhance Confidentiality:** To safeguard the confidentiality of model parameters and effectively mitigate risks associated with model tampering, we incorporate the Trusted Execution Environment based on Intel Software Guard Extensions (SGX). Each participating material research organization is equipped with at least one blockchain node powered by an SGX-enabled processor, responsible for executing model training tasks. During the model aggregation phase, the confidentiality of the aggregated model is verifiable by each node. This setup ensures confidentiality, integrity, and prevents inaccurate model aggregations.

Comment 2.2

Besides the paper does not include a nice discussion section in which one can see the pros and cons of the presented approach.

Response:

Thank you for your insightful feedback regarding the need for a more detailed discussion on the pros and cons of our proposed MatSwarm framework. We acknowledge that the paper would greatly benefit from a clearer exposition of the advantages and limitations associated with our framework.

To enhance the understanding of the implications of our findings and to address the reviewer's comments, we have introduced Section 5.3 "Discussion" in the revised manuscript. This subsection offers a comprehensive discussion of the experimental results, emphasizing the performance advantages and implications for real-world applications of the MatSwarm framework. The discussion is structured around these primary aspects: Security, Usability, Scalability, and Limitation.

5.3 Discussion

5.3.1 Advantages of the MatSwarm framework

- **Security:** *The MatSwarm framework incorporates advanced security measures to ensure data confidentiality and integrity. A key component of our security strategy is the use of Trusted Execution Environments (TEEs), specifically Intel SGX, which protect code and data from external attacks during computation. This approach effectively mitigates poisoning attacks associated with traditional FL setups. Furthermore, our experimental setup included various attack scenarios to test the resilience of the MatSwarm framework. These tests demonstrated that MatSwarm effectively maintains data integrity and model accuracy, even in the face of malicious attempts to corrupt the training process.*
- **Feasibility:** *MatSwarm is crucial for enabling collaborative computation over non-i.i.d. material data, a common challenge due to the diverse nature of data sources and formats in this field. Compared to independent training by organizations and other FL methodologies, our method significantly improves prediction accuracy and generalization ability. This highlights MatSwarm’s potential to unlock the full value of material data, facilitating more informed and accurate materials discovery and development processes. Extensive testing with real-world data from the material science domain validated the usability of the MatSwarm framework. By engaging with actual datasets from participating institutions, we demonstrated the feasibility and accuracy of the models generated through our platform. This use of real data underscores the framework’s ability to address the ‘data silos’ problem prevalent in materials science.*
- **Scalability:** *The MatSwarm framework has been rigorously tested across multiple dimensions, including varying dataset sizes, feature quantities, and participant counts. The results show that the model maintains high and stable predictive accuracy, demonstrating excellent scalability and practical applicability. This consistent performance, even with smaller sample sizes and fewer features, underscores MatSwarm’s capability to adapt to a broad range of scenarios. Such robustness enhances its potential for widespread adoption in collaborative settings that require handling complex, heterogeneous data landscapes. Additionally, the MatSwarm platform utilizes a modular architecture, allowing participants to select appropriate local models and aggregation methods based on their training tasks. As task demands increase, we will continuously iterate and update the platform’s local models and aggregation methods.*

This approach aims to address various challenges in the material science domain, including performance prediction, material classification, and structural optimization, ultimately creating a versatile collaborative computing platform.

5.3.2 Limitations of the MatSwarm Framework

- **Implementation Complexity:** *While incorporating TEEs enhances security and privacy, it also increases the complexity of system setup and operations, necessitating robust infrastructure and specialized expertise. To mitigate this, we provide detailed platform deployment and configuration documentation in the supplementary materials, which stakeholders can use to deploy new training tasks on this platform.*
- **Potential Latency Issues:** *The decentralized nature of blockchain and remote attestation based on TEEs can introduce delays in model training and aggregation. However, in the field of materials science, real-time requirements for training are not stringent. The minor increase in latency is negligible compared to the benefits of resolving the issue of data silos in material data.*
- **Hardware Dependency:** *Dependence on TEEs, such as Intel SGX, to protect data during computation may limit the applicability of our framework in environments without such hardware support. Nevertheless, our demonstration system offers the option to choose whether to use TEEs to secure the confidentiality of the model aggregation process. Even without TEE protection, data security during transmission is ensured through data encryption and secure communication channels. In the future, we plan to offer additional privacy protection technologies, such as homomorphic encryption and differential privacy, to support a broader range of application scenarios.*

In conclusion, the *MatSwarm* framework represents a practically viable solution for addressing the complexities of secure, scalable, and effective data sharing and model training in the materials science domain. By tackling key challenges such as security, usability, and scalability, *MatSwarm* establishes a new benchmark for swarm learning applications in environments with sensitive and heterogeneous data. The addition of Section 5.3 in our manuscript provides readers with a detailed understanding of these aspects, highlighting the comprehensive benefits and potential limitations of our approach in a balanced and transparent manner. This discussion aims to enrich the discourse in this evolving field, offering valuable insights for researchers and practitioners alike.

Comment 2.3

The paper presents the results on a datasets but is difficult to reproduce the research.

Response:

Thank you for your constructive feedback on our manuscript. We appreciate your concerns regarding the replicability of the *MatSwarm* framework as described in our document.

Replicability and Availability of *MatSwarm*: Regarding the replicability of our framework, we understand the importance of making scientific research accessible and reproducible. As mentioned by the reviewers, the original intention of the NMDMS platform construction is to accelerate material sharing and improve the flow and sharing efficiency of data in the field. The *MatSwarm* framework proposed in this paper aims to better serve the material community and enhance data sharing. However, the *MatSwarm* framework was developed as part of the Material Genome Engineering (MGE) project, which involves sensitive data and is subject to national security and confidentiality requirements. Consequently, we are unable to provide direct access to the complete source code of the *MatSwarm* framework.

To address this and still support the scientific community, we have developed a demonstration system that integrates the core functionalities of the *MatSwarm* framework, which can be accessed at <https://github.com/SICC-Group/MatSwarm.git>. Detailed operational procedures can be found in the supplementary videos. This demo system is designed to allow researchers to replicate key aspects of our work and explore the capabilities of the framework under conditions that adhere to the security protocols required by the MGE project. For data access, *MatSwarm* is built upon the NMDMS platform, which is open to the community, and any registered user can access the open data on the platform (currently more than 14 million pieces). Sensitive or non-public data is not directly available, and the *MatSwarm* framework's original intention is to facilitate the use of such data under secure conditions.

Commitment to FAIR Principles: In line with our commitment to the FAIR principles—making our data Findable, Accessible, Interoperable, and Reusable—we are providing detailed documentation and step-by-step guides within the demo system. These resources are crafted to ensure researchers can effectively use the system and independently verify our claims. The supplementary information includes:

1. *Code Availability:* We make the complete source code of *MatSwarm* available on a public repository, ensuring it is accessible to all researchers. The repository includes detailed documentation on setting up and running the code.
2. *Detailed Replication Steps:* The public repository provided in our manuscript now

includes a supplementary section detailing step-by-step instructions on how to replicate the experiments and use the *MatSwarm* framework. This will ensure that researchers can independently verify our claims.

3. *Demonstration Video*: We have provided a supplementary video demonstration of the *MatSwarm* framework in action, showcasing its application in predicting the formation energy of perovskite materials. This visual aid will help researchers better understand the framework's functionality and integration.

We believe these adjustments effectively address the reviewers' concerns and enhance the manuscript's research and scientific value by ensuring transparency and reproducibility within the constraints imposed by security requirements.

Comment 2.4

On the other hand, the framework that has been presented by the author in this case for federated learning is very similar to the one presented in the paper entitled: S-MBDA: A Blockchain-Based Architecture for Secure Storage and Sharing of Material Big-Data by the same authors.

Response:

Thank you for your observations regarding the similarity between the frameworks presented in our papers. While there are foundational technologies shared between the two—particularly blockchain—the focus and applications of these frameworks are distinctly different, catering to different aspects of data management and computation in the materials science domain.

Framework and Focus Differences: The S-MBDA framework primarily emphasizes a secure platform for sharing big data in materials science, utilizing blockchain technology to address challenges in storage, retrieval, and collaborative computation. In S-MBDA, federated learning is implemented as a straightforward case study to demonstrate collaborative computing on the blockchain-based platform. This example uses a basic federated learning approach without modifications to the collaborative computation methods, thus not addressing the heterogeneity issues across different institutional material data.

Conversely, the *MatSwarm* framework represents a significant evolution from the concepts introduced in S-MBDA. *MatSwarm* integrates Trusted Execution Environments (TEEs) alongside blockchain to enhance security during swarm learning processes. This integration ensures the confidentiality and integrity of the aggregation process, a security aspect that was not covered by S-MBDA.

Differences in Addressed Problems: S-MBDA's experiments mainly validate the efficiency and security of storage and retrieval within a blockchain-based infrastructure

for material data. In contrast, *MatSwarm* focuses on the feasibility and accuracy of collaborative computational methods for non-independent and identically distributed (non-i.i.d.) material data under secure conditions. *MatSwarm* introduces an innovative trusted swarm transfer learning method, which allows the global model to learn from the features of local models from various organizations, leading to an optimal global solution. This is further supported by our demonstration system, which offers flexible, modular configuration options allowing users to select different local model algorithms and aggregation methods for their specific training tasks. This adaptability facilitates ongoing iterations and updates to handle increasingly complex material domain training tasks.

Performance Testing Across Multiple Dimensions: We conducted extensive performance testing on security, prediction accuracy, and response times in *MatSwarm*, comparing it against independent organizational training, centralized federated learning, and other existing federated learning solutions. The results showcase the performance advantages of *MatSwarm*. The demonstration system also provides insights into the practical feasibility of the *MatSwarm* framework in real-world scenarios, as detailed in Video 3 of the supplementary materials.

These distinctions underline that while both frameworks utilize blockchain, the advancement and specialization of *MatSwarm* significantly extend its capabilities and applications, specifically tailored for addressing the complex challenges of federated learning in materials science. Additionally, we have explained the difference and connection between the current research work and the previous research work mentioned by the reviewer in our revised paper. We cited the reference of the previous research work and added the following content in the paper:

To accelerate materials science research and development, building on the Materials Genome Engineering (MGE) project [75], we developed the NMDMS platform [65,66] to facilitate the collection, storage, retrieval, and computation of material data. As the cornerstone of MGE’s data applications, NMDMS platform² provides data consumers with access to an extensive repository of material data contributed by over thirty research institutions across China. This platform also serves as a data exchange and sharing hub for materials researchers. Although the NMDMS platform provides basic collaborative computing services, it lacks solutions for handling the inherent limitations of FL in the context of material science. For example, while it achieves relatively high prediction accuracy for i.i.d. (independent and identically distributed) training sets, it falls short in generalization capability for non-i.i.d. training sets and cannot ensure the confidentiality and integrity of parameters during the training process.

²Detailed descriptions of NMDMS are provided in the Supplementary Materials.

Comment 2.5

On the other hand, the lack of review of related works in the last year in such a rapid evolving field together with the lack of conclusions and discussion on the results presented.

Response:

Thank you for your constructive feedback regarding the review of related works and the discussion on our results. We appreciate the opportunity to enhance the quality and completeness of our manuscript.

Review of Recent Related Works: We acknowledge the importance of including a comprehensive review of recent developments in this rapidly evolving field. To address this, we have expanded the related work section to incorporate significant advancements and relevant studies published in the last year. This will ensure that our paper reflects the current state of the art and provides a thorough context for our contributions. In addition, we have included an overview of existing federated learning solutions for addressing non-i.i.d. data issues in *Section 2.1 of the Related Work*. This comprehensive research has laid a solid foundation for the development of *MatSwarm*.

Conclusions and Discussion on Results: We also recognize the need for a more detailed discussion and conclusions section to better articulate the implications of our findings. In the revised manuscript, we have provided a detailed analysis of the results, discussing how our framework improves upon existing methods in terms of security, efficiency, and scalability. We have critically assessed the strengths and limitations of our approach and discussed potential areas for future research. This discussion aims to provide a clear understanding of the impact and relevance of our findings within the broader context of materials science and data security. The revised content is detailed in Comment 2.2.

By addressing these aspects, we aim to enhance the manuscript's contribution to the field and ensure that it provides a valuable reference for researchers and practitioners alike. We thank you for your suggestions, which will undoubtedly help in improving the rigor and depth of our study.

REVIEWER COMMENTS

Reviewer #1 (Remarks to the Author):

The reply from the authors to my (and the other reviewer's) comments display a massive amount of dedicated work. Now the intent and potential of the (renamed) platform is much more clear. In particular, I praise the effort to provide a minimal version of the platform for the reproduction of the published results.

Personally, I have still issues on data points/sets that are not completely public (with all their metadata), but at least the introduced approach ensures that data cannot be tampered (or, very hard to) and may one day be made public.

In any case, the manuscript and supplementary information are clear and for what I can see honest and complete, so I think scientific standards for publication are fully matched.

I recommend publication of the manuscript in its present form.

Reviewer #2 (Remarks to the Author):

Authors have really improved the manuscript after the review process taking into account the reviewers' comments. However, the main issue, that of innovation, behind the paper remains. The proposed method relies on Federated learning and blockchain concepts to deal with a problem in the field of materials.

Authors have claimed innovation based on the features of these data but these features could be shared by other datasets and consequently the method could be used not only for the project mentioned in the paper but for other datasets.

Besides all that is presented is too customized to the project and in fact the sw developed has been integrated on the platform of the mentioned project. No doubt that the manuscripts presents an interesting project with an interesting development that has produced interesting results. But the research aspects and how this solution can be extendable to other domains or even in the material domain remains unclear.

Authors should present the solution not so tighted to the proposed project but present the features of the data if this is the case and then the challenges and present the method as a generalizable solution rather than a solution to a particular problem.

Reviewer #2 (Remarks on code availability):

I have reviewed and checked the git to see the available materials but i have not gone in depth into code or installation.

it is true that now authors have given access to this code but as said by them this is not the same as the one in the project.

Dear Editor and Reviewers:

The authors appreciate your valuable comments. We found them very helpful in revising the paper. We have carefully examined these comments and made appropriate changes to the revised paper. Here we have outlined the details of our responses to the specific comments from the reviewers.

1 Responses to Reviewer 1

Comment 1.1

The reply from the authors to my (and the other reviewer's) comments display a massive amount of dedicated work. Now the intent and potential of the (renamed) platform is much more clear. In particular, I praise the effort to provide a minimal version of the platform for the reproduction of the published results.

Personally, I have still issues on data points/sets that are not completely public (with all their metadata), but at least the introduced approach ensures that data cannot be tampered (or, very very hard to) and may one day be made public.

In any case, the manuscript and supplementary information are clear and for what I can see honest and complete, so I think scientific standards for publication are fully matched.

I recommend publication of the manuscript in its present form.

Response:

Thank you for your thorough review and for the positive feedback regarding our revised manuscript. We appreciate your recognition of the efforts we have put into clarifying the intent and potential of the *MatSwarm* platform. Your comments have been invaluable in guiding our revisions and improving the overall quality of our work.

We are delighted to hear that our responses and revisions have successfully clarified the intent and potential of the *MatSwarm* platform. We also understand your concerns regarding the non-public nature of some data points and datasets. The main starting point of our work is how to maximize the application value of data under the premise that material research and development users choose not to disclose the original data for commercial value or security and privacy issues. We appreciate your acknowledgment that our approach provides strong data tampering prevention and hope that future developments will enable more open data sharing.

Once again, thank you for your valuable feedback and for the positive evaluation of our work. Your insights have been instrumental in refining our manuscript. We look forward to the opportunity to contribute to ongoing research and discussions in the field.

2 Responses to Reviewer 2

Comment 2.1

Authors have really improved the manuscript after the review process taking into account the reviewers comments. However, the main issue, that of innovation, behind the paper remains.

The proposed method relies on Federated learning and blockchain concepts to deal with a problem in the field of materials.

Authors have claimed innovation based on the features of these data but these features could be shared by other datasets and consequently the method could be used not only for the project mentioned in the paper but for other datasets.

Response:

Thank you for your valuable feedback on our manuscript. We appreciate your thoughtful comments and are pleased to hear that you recognize the improvements we have made in response to the previous review process. Our team has worked diligently to address all reviewer comments, ensuring that our work is clearly presented and thoroughly evaluated. We believe these revisions have strengthened the manuscript significantly. Below, we address your concerns regarding the innovation of our method and its adaptability to other datasets.

In our paper, we have detailed that our framework is primarily designed to address challenges in the materials science domain, specifically focusing on achieving collaborative computation among organizations when raw data cannot be shared. For multi-source heterogeneous materials data, the proposed swarm transfer learning method can effectively enhance model accuracy. Furthermore, in other domains with similar collaborative computation requirements without sharing raw data, our framework can serve as a valuable reference. It can be adapted and customized based on specific task requirements to handle datasets from different domains and solve analogous problems.

We acknowledge and appreciate your insight that our method can be applied to other datasets beyond the perovskite formation energy example presented in our manuscript. While this dataset serves as an illustrative example, our framework is applicable to other datasets with various application needs within the materials domain. Regarding the applicability of datasets, we have revised the contributions and discussion sections to clarify that our framework is not limited to the perovskite dataset. Instead, during each shared task, organizations select datasets for training based on task requirements. These datasets often exhibit non-independent and identically distributed (non-i.i.d.) characteristics due to differences in equipment, environment, and technology used for data collection across organizations. To address this, we employ swarm transfer learning to enhance the model accuracy of collaborative computations involving materials data.

Additionally, to accommodate a variety of application scenarios within the materials domain, we offer a range of local models and global aggregation methods. Task initiators can choose appropriate local models and aggregation methods to suit their specific task requirements. The revised contributions are as follows:

1) Innovative Big-Data Security Collaborative Computing Framework for the Material Science Domain: We propose *MatSwarm*, a decentralized collaborative computing framework tailored for the materials field, addressing the collaborative training needs for sensitive datasets. To the best of our knowledge, the application of the *MatSwarm* framework in the materials field is unprecedented. We utilized real datasets from the National Material Data Management and Services (NMDMS) platform to verify its applicability and performance advantages. *Notably, although developed and tested on the NMDMS platform, the MatSwarm framework's design principles and technical implementations offer valuable references for addressing similar challenges in other domains.*

We have expanded the discussion section of the manuscript to further elaborate on this adaptability and its implications for other datasets. The revised discussion provides a more comprehensive view of how *MatSwarm* can be utilized across different datasets and scenarios, enhancing the framework's adaptability and impact. The revised discussion is as follows:

***Adaptability:** MatSwarm is a secure collaborative computing framework designed for non-public data across organizations on the NMDMS, specifically addressing key regression challenges in the materials science domain. In this paper, we demonstrate the capabilities of the MatSwarm framework by using it to predict perovskite formation energies, selecting a perovskite dataset as our example case. To be noted, our framework is suitable for general regression tasks within the materials science domain, such as predicting the elastic properties of silicon materials and optimizing the microstructure of high-performance alloys. For each shared task, participants can choose relevant datasets from their organization based on the task's requirements. This ensures that the framework is not restricted to specific datasets during implementation; instead, it dynamically adapts to select appropriate local datasets according to the specific needs of each task.*

Moreover, although MatSwarm is specifically designed for collaborative computing in the materials science domain, its design principles can be leveraged by other domains with similar needs to construct their own swarm-based collaborative computing frameworks. For other domains with similar application requirements, the framework can be adapted by modifying the objective

function and selecting suitable local models and aggregation methods to fit specific needs. Additionally, in Section 6 of the Supplementary Materials, we provide a detailed guide on how to extend and apply the MatSwarm framework to other domains.

Thank you again for your insightful comments and recognition of our efforts. We are confident that the revisions made have addressed your concerns and have emphasized the innovation and broad adaptability of our work.

Comment 2.2

Besides all that is presented is too customized to the project and in fact the sw developed has been integrated on the platform of the mentioned project. No doubt that the manuscripts presents an interesting project with an interesting development that has produced interesting results. But the research aspects and how this solution can be extendable to other domains or even in the material domain remains unclear.

Authors should present the solution not so tighted to the proposed project but present the features of the data if this is the case and then the challenges and present the method as a generalizable solution rather than a solution to a particular problem.

Response:

Thank you for your insightful comments and for recognizing our efforts in improving the manuscript. We appreciate your feedback, which has been instrumental in guiding the revisions and enhancing the overall quality and clarity of our work. Below, we address your concerns regarding the innovation and generalizability of our solution.

1. Further Explanation on the Innovation of Our Work

The NMDMS platform has successfully aggregated **over 14 million** effective material raw data entries from **more than 30** research institutions across China. However, due to confidentiality or commercial requirements, an additional considerable amount of valuable datasets has only been registered without sharing the original data. This limitation restricts the full mining and analysis of these datasets, thereby hindering the realization of their potential value. To address this challenge, it is crucial to explore methodologies that can fully utilize these registered but non-shared data. By leveraging advanced data-sharing frameworks and secure collaborative computing technologies, we aim to maximize the functionality and impact of the NMDMS platform. This paper focuses on developing and implementing such methodologies in the materials domain, ensuring that even non-public data can contribute to scientific advancements while maintaining necessary confidentiality and commercial protections.

To address this limitation and fully unlock the potential of material data while accelerating the development of new materials, we have developed the *MatSwarm* framework. This framework enhances the existing platform by providing collaborative computing services. By integrating Trusted Execution Environments (TEE) and Swarm Transfer Learning, we not only ensure the security of raw data and the collaborative computing process but also significantly improve the accuracy of model training on heterogeneous material data from multiple sources. This results in a substantial improvement in accuracy compared to independent training by individual organizations.

The introduction of the *MatSwarm* framework is pivotal for the materials field, as it allows for the comprehensive extraction and analysis of material data, thereby maximizing their value and significantly advancing material research and development.

2. Further Explanation on the Generalizability of Our Work

We understand your concerns about the perceived customization of our solution to the specific project and the need for clarity regarding its generalizability across different domains. In response, we have revised the manuscript, particularly focusing on ***Section 6 of Supplementary Materials***, to emphasize the adaptability and broader applicability of our framework. Our revisions highlight that:

Framework Adaptability: The *MatSwarm* framework is particularly well-suited for domains that require collaborative computation and data-driven insights, especially in scenarios where there is a strict requirement for the protection of sensitive data. Furthermore, effective material development often requires multi-institutional collaboration, which *MatSwarm* facilitates by enabling swarm learning. This allows institutions to collaboratively build a shared model, benefiting from pooled data while respecting each organization's data privacy. Domains beyond materials science that require both sensitive data protection and collaborative computation can reference the design and implementation methods of the *MatSwarm* framework.

Generalization Within the Materials Science Domain: In the materials science domain, *MatSwarm* demonstrates its adaptability through the application of consistent methodologies across various datasets and scientific problems. *MatSwarm* can effectively tackle diverse regression problems. As illustrated by the task list in Figure 10 of the supplementary materials, our approach is not limited to predicting the formation energy of perovskites; it can also be applied to predict the elasticity of silicon materials and optimize the microstructure of high-performance alloys, among other regression tasks. Since the execution process of different shared tasks within the *MatSwarm* framework does not significantly differ, we have focused our experimental tests and analysis in the paper primarily on the example of perovskite formation energy. This adaptability allows the framework to provide generalized solutions within materials science and offers a reference for extending its application to other domains.

Extension to Other Domains: Although *MatSwarm* is tailored for the materials science domain, its data-driven methodologies provide a valuable template for addressing

similar challenges in other domains. By leveraging its robust modular design principles, organizations in these domains can customize core components of *MatSwarm*—including the objective function, dataset selection, local models, aggregation methods, and output results—to address challenges similar to those found in materials science. This modular approach allows the framework to be adapted to meet specific application requirements. Furthermore, we provide detailed guidance on customizing the framework’s core components in Section 6 of the Supplementary Materials.

The revised sections demonstrate that *MatSwarm* is not a solution confined to a particular dataset or problem but a versatile and adaptable framework in the materials science domain. Furthermore, the framework can offer valuable references for addressing similar challenges in other domains. *Additionally, we have provided the code for the core functionalities of the framework, along with detailed instructions for code replication. We have also included a demonstration video to illustrate the operation process. These resources serve as a foundational guide for applying and adapting the framework to other domains, offering detailed guidance and demonstrating its feasibility for transfer to other applications.*

Thank you again for your valuable feedback. We hope that the revised content addresses your concerns regarding the innovation and generalizability of our solution.

Comment 2.3

I have reviewed and checked the git to see the available materials but i have not gone in depth into code or installation.

it is true that now authors have given access to this code but as said by them this is not the same as the one in the project.

Response:

Thank you for your feedback and for taking the time to review the materials we have provided. We appreciate your acknowledgment of the access to the code. Our intention in sharing this code was to offer a version of the *MatSwarm* framework that allows for the reproduction of key results and provides insights into the methodology we have developed.

The code available in the repository serves as a minimal version that captures the essential components and functionalities of our framework, although it is not identical to the implementation used in the NMDMS project. This version is designed to demonstrate the framework’s capabilities and facilitate understanding of our approach. The differences between the shared code and the project-specific implementation are primarily due to confidentiality requirements associated with the original project. Certain elements of the project code are proprietary and subject to secure restrictions that prevent us from sharing them publicly. As a result, the shared code has been adjusted to

maintain these confidentiality standards while still demonstrating the core functionalities and methodologies of the *MatSwarm* framework.

We hope that the shared version of the code will be a valuable resource for understanding our framework and inspire further research and development in this area.

REVIEWERS' COMMENTS

Reviewer #2 (Remarks to the Author):

I appreciate the efforts made by the authors to address the previous comments of both reviewers and improve the paper. The enhancements made to the manuscript, significantly strengthen the paper's contribution to the field.

The inclusion of the dedicated platform for this research not only enhances its applicability but also sets a commendable standard for transparency and reproducibility in scientific research.

I am now pleased to recommend the manuscript for publication.

Dear Editor and Reviewers:

The authors appreciate your valuable comments. We found them very helpful in revising the paper. We have carefully examined these comments and made appropriate changes to the revised paper. Here we have outlined the details of our responses to the specific comments from the reviewers.

1 Responses to Reviewer 2

Comment 1.1

I appreciate the efforts made by the authors to address the previous comments of both reviewers and improve the paper. The enhancements made to the manuscript, significantly strengthen the paper's contribution to the field.

The inclusion of the dedicated platform for this research not only enhances its applicability but also sets a commendable standard for transparency and reproducibility in scientific research.

I am now pleased to recommend the manuscript for publication.

Response:

Thank you for your thorough review and for the positive feedback regarding our revised manuscript. We appreciate your recognition of the efforts we have put into clarifying the intent and potential of the MatSwarm. Your comments have been invaluable in guiding our revisions and improving the overall quality of our work. We look forward to the opportunity to contribute to ongoing research and discussions in the field.